# Gaussian Mixture Convolution Networks

**Adam Celarek**[†]**, Pedro Hermosilla**[‡]**, Bernhard Kerbl**[†]**, Timo Ropinski**[‡]**, Michael Wimmer**[†]
[†]TU Wien
[‡]Ulm University

## Abstract

This paper proposes a novel method for deep learning based on the analytical convolution of multidimensional Gaussian mixtures. In contrast to tensors, these do not suffer from the curse of dimensionality and allow for a compact representation, as data is only stored where details exist. Convolution kernels and data are Gaussian mixtures with unconstrained weights, positions, and covariance matrices. Similar to discrete convolutional networks, each convolution step produces several feature channels, represented by independent Gaussian mixtures. Since traditional transfer functions like ReLUs do not produce Gaussian mixtures, we propose using a fitting of these functions instead. This fitting step also acts as a pooling layer if the number of Gaussian components is reduced appropriately. We demonstrate that networks based on this architecture reach competitive accuracy on Gaussian mixtures fitted to the MNIST and ModelNet data sets.

## 1 Introduction

Convolutional neural networks (CNNs) have led to the widespread adoption of deep learning in many disciplines. They operate on grids of discrete data samples, i.e., vectors, matrices, and tensors, which are structured representations of data. While this is approach is sufficiently efficient in one and two dimensions, it suffers the curse of dimensionality: The amount of data is $O(d^k)$, where $d$ is the resolution of data, and $k$ is the dimensionality. The exponential growth permeates all layers of a CNN, which must be scaled appropriately to learn a sufficient number of $k$-dimensional features. In three or more dimensions, the implied memory requirements thus quickly become intractable (Maturana & Scherer, 2015; Zhirong Wu et al., 2015), leading researchers to propose specialized CNN architectures as a trade-off between mathematical exactness and performance (Riegler et al., 2017; Wang et al., 2017). In this work, we propose a novel deep learning architecture based on the analytical convolution of Gaussian mixtures (GMs). While maintaining the elegance of conventional CNNs, our architecture does not suffer from the curse of dimensionality and is therefore well-suited for application to higher-dimensional problems.

GMs are inherently unstructured and sparse, in the sense that no storage is required to represent empty data regions. In contrast to discretized $k$-dimensional volumes, this allows for a more compact representation of data across dimensions, preventing exponential memory requirements. In this regard, they are similar to point clouds (Qi et al., 2017a). However, GMs also encode the notion of a spatial extent. The fidelity of the representation directly depends on the number of Gaussians in the mixture, which means that, given an adequate fitting algorithm and sufficient resources, it can be arbitrarily adjusted. Hence, GMs allow trading representation accuracy for memory requirements on a more fine-grained level than voxel grids.

In this work, we present our novel architecture, the *Gaussian mixture convolution network* (GMCN), as an alternative to conventional CNNs. An important distinguishing feature of GMCNs is that both, data and learnable kernels, are represented using mixtures of Gaussian functions. We use Gaussians with unconstrained weights, positions, and covariances, enabling the mixtures to adapt more accurately to the shape of data than a grid of discrete samples could. Unlike probability models, our approach supports negative weights, which means that arbitrary data can be fitted.

At their core, CNNs are deep neural networks, where each pixel or voxel is a neuron with distinct connections to other neurons, with shared weights and biases. On the other hand, GMCNs represent

data as functions. Hence there are no distinct neurons, and the concept of connection does not exist. Our architecture could thus be more appropriately classified as a *deep functional network*.

Like conventional (discrete) CNNs, our proposed GM learning approach supports several convolution layers, an increasing number of feature channels, and pooling, in order to learn and detect aspects of the input. For the convolution of two mixtures, a closed-form solution is used, which results in a new mixture. Conventional transfer functions like ReLUs do not produce a GM. However, a GM is required to feed the next convolution layer in a deep network. Therefore, we propose to *fit* a GM to the result of the transfer function. The fitting process can simultaneously act as a pooling layer if the number of fitted components is reduced appropriately in successive convolution layers. Unlike most point convolutional networks, which have per-point feature vectors, the feature channels of GMCNs are not restricted to share the same positions or covariance matrices. Compared to conventional CNNs, GMCNs exhibit a very compact theoretical memory footprint in 3 and more dimensions (derivation and examples are included in Appendix A of this paper).

With the presentation of GMCNs, we thus provide the following scientific contributions:

- A deep learning architecture based on the analytical convolution of Gaussian mixtures.
- A heuristic for fitting a Gaussian mixture to the result of a transfer function.
- A fitting method to both reduce the number of Gaussians and act as a pooling layer.
- A thorough evaluation of GMCNs, including ablation studies.

## 2 RELATED WORK

CNNs have been successfully applied to images (e.g., Krizhevsky et al. (2012)) and small 3D voxel grids (e.g., Maturana & Scherer (2015), and Ji et al. (2013)). However, it is not trivial to apply discrete convolution to large 3- or more dimensional problems due to the dimensional explosion. Riegler et al. (2017) reduce memory consumption by storing data in an octree, but the underlying problem remains.

GMs have been used in 3D computer graphics, e.g., by Jakob et al. (2011) for approximating volumetric data, and by Preiner et al. (2014) and Eckart et al. (2016a) for a probabilistic model of point clouds. In particular, the work by Eckart et al. can be used to fit GMs to point clouds, which can then be used as input to our network (see ablation study in Section 5.3). We use the expectation-maximization (EM) algorithm proposed by Dempster et al. (1977) to fit input data, and EM equations for fitting small GMs to larger ones by Vasconcelos & Lippman (1999).

We refer to Ahmed et al. (2018) for a general overview of deep learning on 3D data. Like point clouds, GMs are unordered but with additional attributes (weight and covariance). These additional attributes could be processed like feature vectors attached to points. Therefore, prior work on point clouds is relevant, whereby Guo et al. (2019) provide a good overview.

Several authors propose point convolutional networks, e.g., Li et al. (2018b), Xu et al. (2018), Hermosilla et al. (2018), Thomas et al. (2019), Boulch (2019), and Atzmon et al. (2018). The one by Atzmon et al. is most similar to ours. They add a Gaussian with uniform variance for every point, creating a mixture that is a coarse fitting of the point cloud. After analytical convolution, the result is sampled at the original locations of the point cloud. However, information away from these points is lost. Effectively, such a restricted convolution creates a weighting function for neighboring points.

Radial Basis Function Networks also learn a GM to detect patterns on the input data (Broomhead & Lowe, 1988). However, the goal of such networks differs from our approach. While their goal is, similar to MLPs, to detect patterns on a set of data points, our method uses radial basis functions to detect patterns using a convolution operator on another set of radial basis functions.

## 3 GAUSSIAN MIXTURES AND CONVOLUTION IN MULTIPLE DIMENSIONS

A multidimensional Gaussian function is defined as

$$g(\boldsymbol{x}, \boldsymbol{b}, \boldsymbol{C}) = \frac{1}{\sqrt{(2\pi)^k \det(\boldsymbol{C})}} e^{-\frac{1}{2}(\boldsymbol{x}-\boldsymbol{b})^T \boldsymbol{C}^{-1}(\boldsymbol{x}-\boldsymbol{b})}, \tag{1}$$

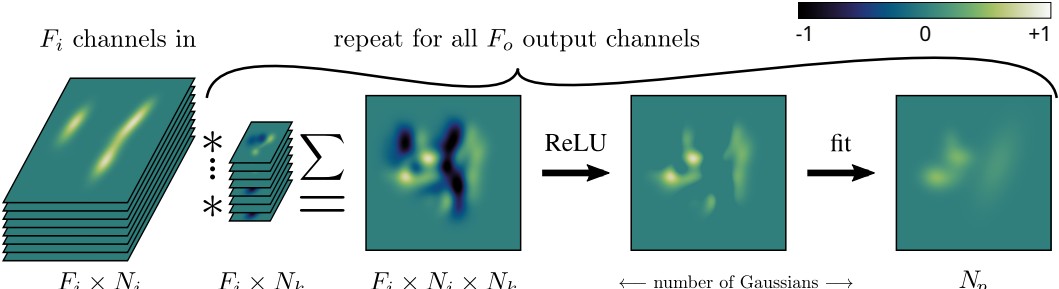

$F_i$ channels in          repeat for all $F_o$ output channels

$F_i \times N_i$     $F_i \times N_k$     $F_i \times N_i \times N_k$     $\longleftarrow$ number of Gaussians $\longrightarrow$     $N_p$

**Figure 1:** Design of a Gaussian convolution layer: The inputs are $F_i$ channels of GMs with $N_i$ components each, which are analytically convolved with $F_i$ learnable kernels, $N_k$ components each. The convolved mixture contains $F_i \times N_i \times N_k$ components. After applying the ReLU, a new GM with $N_p$ components is fitted. This is repeated for all of the $F_o$ output feature channels. The resulting tensor becomes the input of the next convolution channel, with the number of output channels $F_o$ becoming $F_i$, and $N_p$ becoming $N_i$. The fitting must be fully differentiable with respect to the input of the ReLU. In this example, $F_i = 8, N_i = 16, N_k = 5$, and $N_p = 8$.

where $k$ is the number of dimensions, $\boldsymbol{b}$ the shift or center point, and $\boldsymbol{C}$ the shape (covariance matrix) of the Gaussian. A GM is a weighted sum of multiple such Gaussians:

$$\text{gm}(\boldsymbol{x}, \boldsymbol{a}, \boldsymbol{B}, \mathbf{C}) = \sum_{i=0}^{N} a_i \, \text{g}(\boldsymbol{x}, \boldsymbol{B}_i, \mathbf{C}_i) \tag{2}$$

A GM is not a probabilistic model since the weights $a_i$ can be negative and do not necessarily sum up to $1$. The convolution of two Gaussians according to Vinga (2004) is given by

$$\text{g}(\boldsymbol{x}, \boldsymbol{b}, \boldsymbol{C}) * \text{g}(\boldsymbol{x}, \boldsymbol{\beta}, \boldsymbol{\Gamma}) = \text{g}\left(\boldsymbol{x}, \boldsymbol{b} + \boldsymbol{\beta}, \boldsymbol{C} + \boldsymbol{\Gamma}\right). \tag{3}$$

Due to the distributive property of convolution, we have:

$$\sum_{n=0}^{N} \text{g}_n(\boldsymbol{x}) * \sum_{m=0}^{M} \text{g}_m(\boldsymbol{x}) = \sum_{n=0}^{N} \sum_{m=0}^{M} \text{g}_n(\boldsymbol{x}) * \text{g}_m(\boldsymbol{x}), \tag{4}$$

where $\text{g}_i(\boldsymbol{x})$ is a short hand for $a_i \, \text{g}(\boldsymbol{x}, \boldsymbol{B}_i, \mathbf{C}_i)$. Clearly, the convolution of one mixture with $N$ terms with another mixture with $M$ terms yields a mixture with $N \times M$ terms.

## 4 GAUSSIAN MIXTURE CONVOLUTION NETWORKS

Conventional CNNs typically consist of several feature channels, convolution layers, a transfer function, and pooling layers. In this section, we will show that all these components can be implemented by GMCNs as well. As a result, our overall architecture is similar to discrete CNNs, except that data and kernels are represented by Gaussian functions rather than pixels or voxels.

**Convolution layer.**    All batch elements and feature channels contain the same number of components, allowing for packing Gaussian data (i.e., weights, positions, and covariances) into a single tensor. Like in conventional CNNs, each output channel ($F_o$) in a GMCN is the sum of all $F_i$ input channels, convolved with individual kernels (Figure 1). In practice, this means that multiple convolution results are concatenated in the convolution layer. This results in a mixture with $N_o = F_i N_i N_k$ components, where $N_i$ and $N_k$ are component counts in input and kernel mixtures, respectively.

**Transfer function.**    The ReLU is a commonly used transfer function, whose ramp-like shape is defined by $\varphi(x) = \max(0, x)$. When applying this transfer function to a mixture gm, we get

$$\varphi(\text{gm}(\boldsymbol{x})) = \max(0, \sum_{n=0}^{N_o} \text{g}_n(\boldsymbol{x})). \tag{5}$$

We can use Equation 5 to evaluate the mixture at particular locations of $\boldsymbol{x}$, as shown in Figure 2b. However, any subsequent convolution layers in a GMCN will again require a GM as input. Another

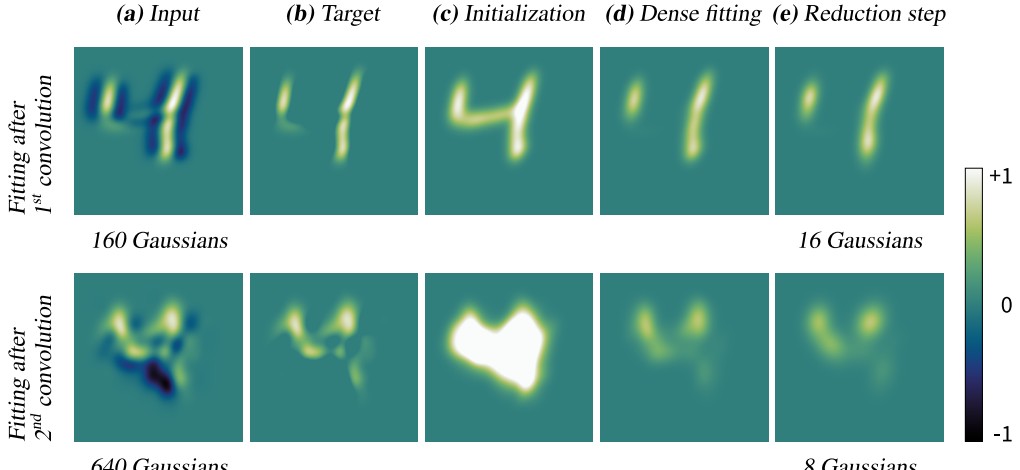

**Figure 2:** Examples of the fitting pipeline described in Section 4.1: The input mixture is shown in (a) and in (b) after applying the ReLU. (c) and (d) show the coarse approximation ($a'$) and final weights ($a''$) of the dense fitting. The mixture after the reduction step is shown in (e). The overall fitting error is bound by dense fitting, as theoretically, the reduction can be omitted altogether (or a very high number of Gaussians used).

problem inherent to the definition of GM convolution in Section 3, is that the number of intermediate output components (and thus implied memory requirements of hidden layers) is set to increase with each convolution. We address both of these problems by introducing a dedicated fitting step that produces a new, smaller mixture to represent the result of $\varphi(\mathrm{gm}(\boldsymbol{x}))$ and act as input for subsequent GMCN layers.

## 4.1 GAUSSIAN MIXTURE FITTING

Fitting a GM to the output of the transfer function is not trivial. In order to propagate the gradient to the kernels, the fitting method must be differentiable with respect to the parameters of the Gaussian components in gm. A naive option is to sample the GM, evaluate $\varphi$, and perform an EM fitting. However, this quickly becomes prohibitive due to the costs associated with sampling several points from each Gaussian, iterations of likelihood computations, and computation of the associated gradients. We decided to first fit a dense GM to the transfer function output (e.g., a ReLU $\varphi(\mathrm{gm}(\boldsymbol{x}))$, shown in Figure 2b), and then reduce the number of Gaussians in a separate step.

**Dense fitting.** To reduce the degrees of freedom that we must optimize for during fitting, we first assume that some Gaussians in the output of the last convolution step are meaningfully placed. Hence, we freeze their positions and adjust only their weights. One possible solution to adjust the GM's weights would be to set up an overdetermined linear system, defined by a set of sample points in the mixture and the corresponding output values. This set of equations could then be fed to a solver for determining the optimal GM weights to fit the outputs according to least squares. However, our experiments have shown that this approach quickly becomes intractable, as it requires too many samples to achieve a fitting that is both stable wrt. their selection and generalizes well to locations that lie between selected points (see Appendix B for details).

To avoid the implied restrictive memory overhead and severe bottleneck it would cause, we chose to pursue a less resource-heavy solution and propose the following heuristic instead: First, a coarse approximation ($\boldsymbol{a'}$) of the new weights is computed according to

$$a'_i = \begin{cases} a_i, & \text{if } a_i > 0 \\ \epsilon, & \text{otherwise} \end{cases}, \tag{6}$$

where $a_i$ are the weights of the old mixture, and $i$ runs through all Gaussians. The new weights $\boldsymbol{a''}$ are then computed using a correction based on the transfer function

$$a''_i = a'_i \times \frac{\varphi\left(\mathrm{gm}(\boldsymbol{B}_i, \boldsymbol{a}, \boldsymbol{B}, \mathbf{C})\right)}{\mathrm{gm}(\boldsymbol{B}_i, \boldsymbol{a'}, \boldsymbol{B}, \mathbf{C})}, \tag{7}$$

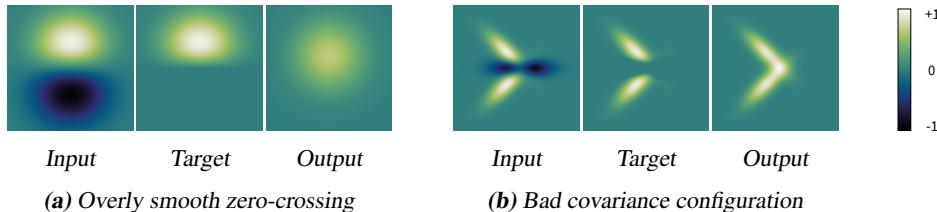

*Input*     *Target*     *Output*         *Input*     *Target*     *Output*

**(a)** *Overly smooth zero-crossing*      **(b)** *Bad covariance configuration*

**Figure 3:** Failure cases for dense fitting: We show inputs, ReLU reference outputs, and our own. In (a), the zero-crossing is overly smooth, and the output weight too small. The positive Gaussian should have been moved up instead. In (b), covariances have a bad configuration, they should have been reduced, and the position moved.

where $B$ contains the positions, $\mathbf{C}$ the covariance matrices, and $\varphi$ represents the ReLU. Effectively, the fraction computes a correction term for the coarse approximation $a'$. It is guaranteed, that the new weights $a''$ are positive, just like we would expect from a ReLU. Examples of the coarse approximation ($a'$) are given in Figure 2c, and the final fitting ($a''$) in Figure 2d. The fitting is most precise at the Gaussian centers. Problems arise at the discontinuity of the ReLU, where the result can be overly smooth, as seen in Figure 3a, or with an adverse configuration of covariances, as is the case in Figure 3b. Appendix C explains some of the design choices in more detail.

**Reduction step.** The next step in our fitting performs expectation-maximization (EM) to reduce the number of Gaussians in the mixture. We implemented two variants, a simpler *modified EM* variant based on the derivations from Vasconcelos & Lippman (1999), and our own tree-based hierarchical EM method. For the simpler, modified EM variant, we select the $N_p$ Gaussians with the largest integrals out of the $N_o$ available ones. Then, one step of EM is performed, balancing accuracy with compute and memory resources, resulting in the fitting. However, this procedure implies $O(N_o \times N_p)$ in terms of memory and computation complexity, rendering it too expensive for larger mixtures.

To remedy this issue, we propose tree-based hierarchical expectation-maximization (TreeHEM). A fundamental idea of TreeHEM is to merge Gaussians in close proximity preferentially. To this end, the Gaussians are first sorted into a tree hierarchy, based on the Morton code of their position, similar to Lauterbach et al. (2009). This tree is traversed bottom-up, collecting up to $2T$ Gaussians and performing a *modified EM* step, which reduces the number to $T$ Gaussians. The fitted Gaussians are stored in tree nodes. The process continues iteratively, collecting again up to $2T$ Gaussians followed by fitting until we reach the root. Finally, the tree is traversed top-down, greedily collecting the Gaussians with most mass from the nodes until the desired number of fitted Gaussians is reached. Hence, TreeHEM yields an $O(N_o \log N_o)$ algorithm. Experiments given in Section 5.3 show, that a value of $T = 2$ works best in terms of computation time, memory consumption, and often fitting error. Examples of the reduction are shown in Figure 2e. For more, details please see Appendix D.

## 4.2 REGULARIZATION AND MAINTAINING VALID KERNELS

In order to add support for regularization to the learning process, we can apply weight decay, which pushes weights towards zero and the covariance matrices towards identity:

$$d_i = a_i^2 + \text{mean}\left((\mathbf{C}_i - I) \odot (\mathbf{C}_i - I)\right), \tag{8}$$

where $d_i$ is the weight decay loss for kernel Gaussian $i$, and $a$, $\mathbf{C}$, and $I$ are the weights, covariances, and the identity matrix, respectively, and $\odot$ is the Hadamard product.

Training the GMCN includes updates to the covariance matrices of kernel Gaussians. In order to prevent singular matrices, $C'$ is learned, and then turned into a symmetrical non-singular matrix by

$$C = {C'}^T C' + I\epsilon. \tag{9}$$

## 4.3 POOLING

In conventional CNNs, pooling merges data elements and reduces the (discrete) domain size at the same time. As filter kernels remain identical in size but are applied to a smaller domain, their receptive field effectively increases.

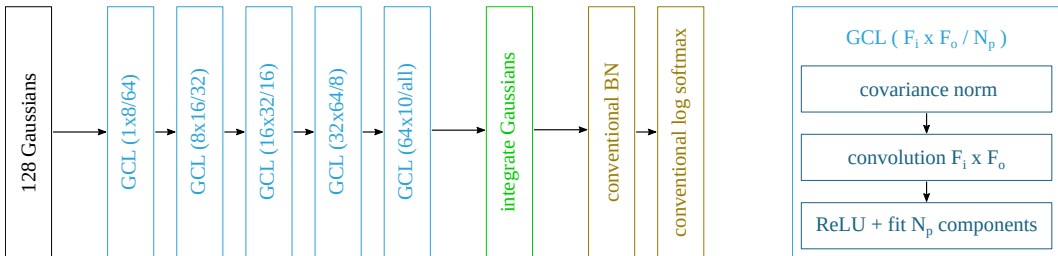

**Figure 4:** The GMCN used in for our evaluation uses 5 Gaussian convolution layers (GCL, see right-hand side), integration of feature channel mixtures, and conventional transfer functions. See Section 5 for more details.

For pooling data with GMCNs, we can set up the fitting step (Section 4.1) such that the number of fitted Gaussians is reduced by half on each convolution layer. While this effectively pools the data Gaussians by increasing their extent, it does not scale the domain. Thus, using the same kernels on the so-fitted data does not automatically lead to an increased receptive field. To achieve this, the kernel Gaussians would have to increase in size to match the increase in the data Gaussians' extent. Instead, similar to conventional CNNs, we simply reduce the data mixtures' domain by scaling positions and covariances. We compute the scaling factor such that after each layer, the average trace of the covariance matrices becomes the number of dimensions (2 or 3 currently), leading to an average variance of 1 per dimension. Effectively, this leads to Gaussian centers moving closer together and covariances becoming smaller. Hence, the receptive field of each kernel becomes larger in relation to the original data scale, even if the kernel size is fixed.

# 5 EVALUATION

To evaluate our architecture, we use the proposed GMCN architecture to train classification networks on a series of well-known tasks. We used an NVIDIA GeForce 2080Ti for training. For all our experiments, we used the same network architecture and training parameters shown in Figure 4. First, we fit a GM to each input 2D or 3D sample by using the k-means algorithm for defining the center of the Gaussians and one step of the EM algorithm to compute the covariance matrices. The mixture weights are then normalized such that the mixtures have the same activation across all training samples on average. Covariance matrices and positions are normalized using the same procedure as for pooling (Section 4.3). The input GM is then processed by four consecutive blocks of Gaussian convolution and ReLU fitting layers. The numbers of feature channels are $[8, 16, 32, 64]$ and the number of data Gaussians is halved in every layer. Finally, an additional convolution and ReLU block outputs one feature channel per class. For performance reasons, we do not reduce the number of Gaussians in this last block. These feature channels are then integrated to generate a scalar value, which is further processed by a conventional batch-normalization layer. The resulting values are converted to probabilities by using the $\mathrm{Softmax}$ operation, from which the negative-log-likelihood loss is computed.

The kernels of each convolutional layer are represented by five different Gaussians, with weights randomly initialized from a normal distribution with variance $1.0$ and mean $0.1$. One Gaussian is centered at the origin, while the centroids of the remaining ones are initialized along the positive and negative x- and y-axes at a distance of $2.5$ units from the origin. For the 3D case, the remaining Gaussians are placed randomly at a distance of $2.5$ units from the origin. The covariance factor matrix of each Gaussian is initialized following $(\boldsymbol{I} + \boldsymbol{R} * 0.05) * 0.7$, where $\boldsymbol{I}$ is the identity, and $\boldsymbol{R}$ a matrix of random values in the range between $-1$ and $1$. The covariance matrix was then computed according to Equation 9. We determined these values using a grid search. Examples of kernels are shown in Appendix E.

All models are trained using the Adam optimizer, with an initial learning rate of $0.001$. The learning rate is reduced by a scheduler once the accuracy plateaus. Moreover, we apply weight decay scaled by $0.1$ of the learning rate to avoid overfitting, as outlined in Section 4.2.

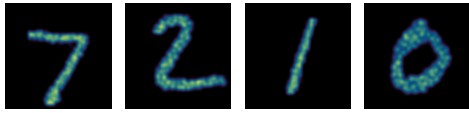

**Figure 5:** GM fitting examples of MNIST digits. 64 Gaussians were used to represent each input sample.

**Table 1:** ModelNet10 classification results of our method in comparison with PointNet and PointNet++. We show two variants of competitor training: 1., on a set of points that matches the memory footprint of the GM, and 2., trained on the same GM inputs. The number of Gaussian (G) and the number of points (P) are indicated in the header for each column.

|  |  | *32 G / 106 P* | *64 G / 213 P* | *128 G / 426 P* | *#Params* |
|---|---|---|---|---|---|
| Points | PointNet | 91.3 % | 92.0 % | 92.2 % | 3.4 M |
|  | PointNet++ | 92.8 % | 93.3 % | 93.9 % | 1.4 M |
| GM | PointNet | 90.7 % | 91.9 % | 91.6 % | 3.4 M |
|  | PointNet++ | 91.8 % | 92.1 % | 92.4 % | 1.4 M |
|  | Ours | 92.1 % | 92.3 % | 93.3 % | 215 K |

## 5.1 2D DATA CLASSIFICATION (MNIST)

The MNIST data set (Lecun et al., 1998) consists of 70 K handwritten digits from 10 different classes. In this task, models have to predict the digit each image represents, where performance is measured as overall accuracy. We trained our GMCN architecture using the standard train/test splits by fitting 64 Gaussians to each image and processing them with our model. Figure 5 shows some examples of the generated GMs. Our trained model achieved an accuracy of 99.4 % on this task, which is close to the 99.8 % currently reported by other state-of-the-art methods (Byerly et al., 2020; Mazzia et al., 2021). However, when compared to a multi-layer perceptron (Simard et al., 2003) (98.4 %) or vanilla CNN architectures (Lecun et al., 1998) (99.2 %), which more closely reflect the GMCN design we used, our method performs better. Compared to point-based methods, our network achieves competitive performance (PointNet (Qi et al., 2017a) 99.2 % and PointNet++ (Qi et al., 2017b) 99.5 %).

## 5.2 3D DATA CLASSIFICATION (MODELNET 10)

ModelNet10 (Zhirong Wu et al., 2015) is a data set used to measure 3D classification accuracy and consists of 3D models of different man-made objects. The data set contains 4.8 K different models from 10 different categories. We trained our network on ModelNet10 using the standard train/test splits without data augmentation. We used the point clouds provided by Qi et al. (2017b), which were obtained by randomly sampling the surface of the 3D models. We then fit 128 Gaussians to these point clouds to generate the input for our network. Our trained model achieved an accuracy of 93.3 %, which is similar to other state-of-the-art classifiers: 94.0 % (Klokov & Lempitsky, 2017) or 95.7 % (Li et al., 2018a). Competing CNN-based methods reach 92% (Maturana & Scherer (2015)) and 91.5% (Riegler et al. (2017)). Appendix F provides additional, more detailed results.

In Table 1, we again compare our method with PointNet (Qi et al., 2017a) and PointNet++ (Qi et al., 2017b), as they represent two well-established methods for point cloud processing. To enable a fair comparison with each point-based technique, we ran two variants: The first uses a point cloud as input, with a number of points equivalent to the memory footprint of our GM representation. The second uses the same GM inputs as our network, where weight and covariance matrix are processed as additional features for each point. Models are trained with various input sizes: 32 Gaussians (106 points), 64 Gaussians (213 points), and 128 Gaussians (426 points). We found that our method outperforms PointNet on all experiments using an order of magnitude fewer parameters. It also achieves a similar performance to the point-based hierarchical architecture PointNet++ using five times fewer parameters. Our method outperforms both architectures when processing GMs directly.

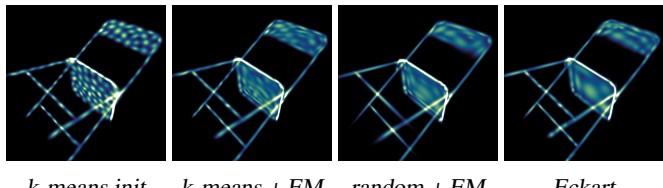

*k-means init*  *k-means + EM*  *random + EM*  *Eckart*

**Figure 6:** GM fitting examples of a ModelNet object with 128 Gaussians for k-means and EM algorithms and 125 Gaussians for Eckart et al. (2016b). The k-means initialization method does not produce smooth surface fittings, but Gaussians have uniform covariances. EM with k-means produces results that are relatively sharp on edges and smooth on surfaces. EM with random initialization and Eckart produces even smoother surfaces.

**Table 2:** Accuracy and training time on the ModelNet10 data set for our model wrt. the number of layers. Our network benefits from additional layers. However, accuracy saturates after five layers. While training times in our proof-of-concept system are not competitive, GMCNs leave much room for optimizations in future work.

|               | 1      | 2      | 3      | 4      | 5      | 6      |
|---------------|--------|--------|--------|--------|--------|--------|
| Accuracy      | 72.5 % | 89.4 % | 91.2 % | 92.0 % | 92.6 % | 92.3 % |
| Training time | 0.5h   | 2h     | 5h     | 10h    | 21h    | 41h    |

## 5.3 ABLATION STUDIES

In this section, we discuss additional experiments carried out to validate the individual elements of our network. For these ablation studies, we use the ModelNet10 data set in all experiments.

**Network depth.** First, we evaluate the network's performance wrt. the depth of our GMCN to determine the optimal number of layers. Table 2 shows the classification accuracy on the ModelNet10 data set for different models with varying numbers of convolution layers. We confirmed that GMCN can benefit from additional layers; however, the accuracy stops increasing once we reach five layers.

**Gaussian mixture fitting.** We evaluated the two proposed fitting algorithms (Section 4.1) used after each convolution layer. Three different reduction variants were tested: modified EM, TreeHEM with $T = 2$ and 4 Gaussians per node. Table 3 shows the results of these methods in terms of the fitting error in the first three layers and the accuracy of the model when using 64 and 128 Gaussians as input, respectively. TreeHEM obtains a lower fitting error in most of the layers when compared to the EM algorithm. Moreover, the tree-based method also requires less memory during training compared to EM, which runs out of memory when trained on inputs represented by 128 Gaussians. However, EM obtains a slightly higher classification accuracy for 64 input Gaussian. Out of the two TreeHEM variants, the $T = 2$ performs better in terms of compute time, memory, fitting error, and classification accuracy. Memory and time are easily explained by the number Gaussians cached in the tree. We infer, that accuracy is reduced in higher $T$ settings due to smearing of Gaussians (i.e., loss of fine details), which is more likely to occur when more of them are merged simultaneously.

**Input data fitting.** Lastly, we evaluate how the model is affected by the different algorithms used to fit the input data. We compare four different methods: k-means initialization plus one single step of

**Table 3:** Evaluation of our proposed fitting algorithm. We find that TreeHEM achieves lower errors than the modified EM algorithm when fitting the output of different layers. Nonetheless, accuracy is similar. Moreover, we can see that the EM algorithm runs out of memory when using 128 input Gaussians and a batch size of 21.

|          | RMSE per Layer (#G=64) | | | # Gaus. | | Memory | | Training time | |
|----------|------|------|------|--------|--------|-------|-------|------|------|
|          | 1    | 2    | 3    | 64     | 128    | 64    | 128   | 64   | 128  |
| EM       | 0.45 | 0.50 | 0.36 | 92.2 % | n/a    | 2.5GB | n/a   | 6h   | n/a  |
| Tree (4) | 0.35 | 0.22 | 0.25 | 91.3 % | 92.9 % | 1.6GB | 6.8GB | 5h   | 22h  |
| Tree (2) | 0.36 | 0.21 | 0.22 | 92.0 % | 93.4 % | 1.3GB | 5.6GB | 4h   | 19h  |

**Table 4:** Accuracy of our model on the ModelNet10 data set for input GM generated using different algorithms. Results show that *k-means* produces the best accuracy, while more advanced methods that generate uneven covariance matrices, such as Eckart et al. (2016b), reduce the classification accuracy of the network.

| *k-means* | *k-means+EM* | *rand+EM* | *Eckart* |
|-----------|--------------|-----------|----------|
| 93.3 % | 92.1 % | 92.8 % | 91.0 % |

EM (*k-means*), k-means initialization plus multiple EM steps (*k-means+EM*), random initialization plus multiple EM steps (*rand+EM*), and the method proposed by Eckart et al. (2016b) (*Eckart*). Figure 6 shows the resulting GMs of the four different algorithms for two different models from ModelNet10. We see that *k-means* produces a set of Gaussians with relatively uniform shapes, but overall does not fit the data accurately. *Random init + EM* and *Eckart* can fit the model with high precision but also produces some blurred areas. *K-means init + EM* achieves the best representation of the four methods.

The test accuracy of our network for the four types of input GMs is reported in Table 4. *K-means* initialization achieves the best accuracy while *Eckart* performs worst. We believe these results can be explained by the uneven Gaussian covariance matrices in methods such as *Eckart* and *rand+EM*.

## 6 DISCUSSION AND CONCLUSION

To the best of our knowledge, we are the first to propose a deep learning method based on the analytical convolution of GMs with unconstrained positions and covariance matrices. We have outlined a complete architecture for the task of general-purpose deep learning, including convolution layers, a transfer function, and pooling. Our evaluation demonstrates that this architecture is capable of obtaining competitive results on well-known data sets. GMCNs provide the necessary prerequisites for avoiding the curse of dimensionality (Appendix A). Hence, we consider our architecture as a promising candidate for applications that require higher-dimensional input or compact networks.

**Limitation and future work.** Currently, the most significant bottleneck to limit the universal application of our work is the—unavoidable—fitting process. As reflected by the results in Table 2, the implied computation time limits the size of GMCNs that can be trained with consumer-grade hardware in an appropriate amount of time. For the same reason, data augmentation was omitted in our tests. The basic architecture presented in Section 5 is insufficient for achieving competitive results on more complex tasks, such as classification on ModelNet40 (details in Appendix F.2) and FashionMNIST (86% accuracy, others more than 90% (Xiao et al., 2017; zalandoresearch, 2019)).

However, early experiments have shown that performance can be drastically improved by embedding a fitting step in the convolution stage. Specifically, we can use an on-the-fly convolution of the positions to construct an auxiliary tree data structure, in a manner similar to Section 4.1. The resulting tree is then used to perform a full convolution and fitting in one stage. We have identified highly parallel algorithms for the required steps and confirmed that applying these optimizations to our methods can significantly reduce memory traffic and improve computation times by more than an order of magnitude.

We further conjecture that the achieved prediction quality with our architecture is currently limited by fitting accuracy (see Tables 3, 10, 11 and Appendix F). This could be addressed by sampling the GM and performing one or two EM steps on these samples. Doing so should become feasible once the number of Gaussians is brought down using the convolution fitting step described above. Presumably, even a slight increase in accuracy would enable us to achieve state-of-the-art results on all tested use cases.

Additional extensions to raise the applicability of GMCNs would be the introduction of transposed convolution, as well as skip connections (Ronneberger et al., 2015). We expect that these tasks could be solved with a reasonable amount of additional effort. We hope to enable novel solutions for 3-dimensional problems with these extended GMCNs in future work, such as hole-filling for incomplete 3D scans or learning physical simulations for smoke and fluids.

## REPRODUCIBILITY

The source code for a proof-of-concept implementation, instructions, and benchmark datasets are provided in our GitHub repository (https://github.com/cg-tuwien/Gaussian-Mixture-Convolution-Networks).

## ACKNOWLEDGMENTS

The authors would like to thank David Madl, Philipp Erler, David Hahn and all other colleagues for insightful discussions, Simon Fraiss for implementing fitting methods for 3d point clouds. We also acknowledge the computational resources provided by TU Wien. This work was in part funded by European Union's Horizon 2020 research and innovation programme under the Marie Skłodowska-Curie grant agreement No 813170, the Research Cluster "Smart Communities and Technologies (Smart CT)" at TU Wien, and the Deutsche Forschungsgemeinschaft (DFG) under grant 391088465 (ProLint).

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

## A  THEORETICAL MINIMAL MEMORY FOOTPRINT (AND PARAMETER COUNT)

This section provides formal equations and concrete examples to outline the theoretical minimum memory footprint of a GMCN and enable assessment of its overall potential for reducing memory consumption. For this, we made the following assumptions:

- None of the modules (convolution/reduction fitting, transfer function fitting, pooling) rely on caches.
- Programming code is not counted.
- Minimal storage requires 6 floats for a 2D Gaussian, and 10 floats for a 3D Gaussian.
- Fitting can be *integrated into the convolution layer* (see the following paragraph) and produces a mixture with a specified number of Gaussians $N_o$. No fitting is performed if the specified number is equal to the number of Gaussians produced by the convolution.

In conventional CNNs, the convolution operation, that is "per-pixel kernel-sized window, element-wise multiplication, and sum", is usually built into one compact routine, or CUDA kernel. Accordingly, intermediate results are never stored in memory. The proof-of-concept implementation described in this paper is not compact in the same way. The closed-form solution of the convolution with GMCNs yields a large number of intermediate output Gaussians, which are then greatly reduced in the fitting step before being forwarded to the next layer. The described prototype explicitly stores these intermediate results, making it easier to implement and explain. As outlined in Chapter 6, the integration of a fitting step directly into the convolution is possible with our approach, albeit more complex compared to conventional CNNs. However, in comparison to conventional CNNs, such a compact implementation of GMCNs can achieve significant savings on memory footprint.

**Integrated convolution-fitting module**  A convolution module has $F_i$ input feature channels and $F_o$ output feature channels, with batch size set to $B$. The memory consumed by the incoming data is accounted towards the previous module. Given incoming data mixtures containing $N_i$ Gaussians and kernels containing $N_k$ Gaussians, our proof-of-concept implementation produces and stores $N_o = F_i N_i N_k$ Gaussians, which will be reduced before pooling is applied. In contrast, an integrated convolution-fitting module can produce a smaller number of Gaussians $N_o$ directly. Ideally, $N_o = 2N_p$, such that the the subsequent pooling layer can select half of the provided inputs to yield $N_p$ output Gaussians.

The minimal memory footprint in terms of Gaussians $(G)$ is the sum of kernels $(K)$, and data $(D)$ times 2 (for the gradient):

$$K = F_i F_o N_k$$
$$D = B(F_o N_o + F_o N_p)$$
$$G = 2(K + D). \tag{10}$$

$K$ can also be used to compute the number of parameters of the given module by multiplying with the representation used (6/10 floats for 2D/3D, respectively).

**Minimal memory footprint for trained model**  The factors and results for the model used for our evaluation (shown in Figure 4) are given in Tables 5 and 6. Overall, if $N_o = 2 \times N_p$ is used, the

**Table 5:** Minimal memory footprint for the network shown in Figure 1, using Equations 10, if $N_o = 2N_p$. The total number of Gaussians for each convolution module is given as $G$. Theoretical memory usage in megabytes is given in columns $M_{2D}$ and $M_{3D}$, using 2D and 3D Gaussians, respectively.

| $B$ | $F_i$ | $F_o$ | $N_i$ | $N_o$ | $N_p$ | $N_k$ | $K$ | $D$ | $G$ | $M_{2D}$ | $M_{3D}$ |
|---|---|---|---|---|---|---|---|---|---|---|---|
| 32 | 1 | 8 | 128 | 128 | 64 | 5 | 40 | 49,152 | 98,384 | 2.25 | 3.75 |
| 32 | 8 | 16 | 64 | 64 | 32 | 5 | 640 | 49,152 | 99,584 | 2.28 | 3.80 |
| 32 | 16 | 32 | 32 | 32 | 16 | 5 | 2,560 | 49,152 | 103,424 | 2.37 | 3.95 |
| 32 | 32 | 64 | 16 | 16 | 8 | 5 | 10,240 | 49,152 | 118,784 | 2.72 | 4.53 |
| 32 | 64 | 10 | 8 | 8 | 4 | 5 | 3,200 | 3,840 | 14,080 | 0.32 | 0.54 |

**Table 6:** Minimal memory footprint for network shown in Figure 1, using Equations 10, if implemented without combined convolution-fitting. The total number of Gaussians for each convolution module is given as $G$. Theoretical memory usage in megabytes is given in columns $M_{2D}$ and $M_{3D}$.

| $B$ | $F_i$ | $F_o$ | $N_i$ | $N_o$ | $N_p$ | $N_k$ | $K$ | $D$ | $G$ | $M_{2D}$ | $M_{3D}$ |
|---|---|---|---|---|---|---|---|---|---|---|---|
| 32 | 1 | 8 | 128 | 640 | 64 | 5 | 40 | 180,224 | 360,528 | 8.25 | 13.75 |
| 32 | 8 | 16 | 64 | 2,560 | 32 | 5 | 640 | 1,327,104 | 2,655,488 | 60.78 | 101.30 |
| 32 | 16 | 32 | 32 | 2,560 | 16 | 5 | 2,560 | 2,637,824 | 5,280,768 | 120.87 | 201.45 |
| 32 | 32 | 64 | 16 | 2,560 | 8 | 5 | 10,240 | 5,259,264 | 10,539,008 | 241.22 | 402.03 |
| 32 | 64 | 10 | 8 | 2,560 | 4 | 5 | 3,200 | 820,480 | 1,647,360 | 37.71 | 62.84 |

model has a theoretical minimal memory footprint of 434,256 total 3D Gaussians, which is less than 17 megabytes. In comparison, the proof-of-concept implementation presented in this paper has a theoretical use of 781 megabytes. In practice, it still requires several times more, since it uses convenience data structures that were not exhaustively optimized.

## B  COMPARISON OF LEAST-SQUARES SOLUTION AND HEURISTIC FOR ReLU FITTING

In this section, we will compare a least-squares solution with our heuristic presented in Section 4.1 for the fitting of a transfer function.

The least-squares solution has to solve the following problem

$$A\boldsymbol{y} = \boldsymbol{t}, \tag{11}$$

where $\boldsymbol{t}_i = \varphi(\mathrm{gm}(D_i))$ are the target values at positions $D$, and $A$ is a matrix:

$$A = \begin{pmatrix} \gamma_0(D_0) & \dots & \gamma_N(D_0) \\ \vdots & \ddots & \vdots \\ \gamma_0(D_N) & \dots & \gamma_N(D_N) \end{pmatrix}, \tag{12}$$

with $\gamma_i(\boldsymbol{x}) = \frac{1}{\sqrt{(2\pi)^k \det(C)}} e^{-\frac{1}{2}(\boldsymbol{x}-B_i)^T C_i^{-1}(\boldsymbol{x}-B_i)}$. $B_i$ is the centroid, $C_i$ the covariance of Gaussian $i$, $N$ is the number of Gaussians, and $k$ the number of dimensions (e.g., 2 or 3). In other words, A contains the evaluations of the mixture Gaussians with the weight set to one, on a set of points contained in $D$.

We evaluated several choices for $D$:

- $D = B$, evaluate the mixture at the centre points of its Gaussians
- $D = rand(M, k)$, positions within the domain
- $D = concat(B, rand(M, k))$, mixture centres and random positions within the domain

Figure 7 shows part of our test input, along with the result of our heuristic. After seeing the results in Figure 8 and timings in Table 7, we refrained from computing the RMSE and selected the heuristic for our network. The least-squares solution overshoots away from the sampling points, and the compute resources are several magnitudes higher compared to our heuristic.

The matrix A with choice $D = B$ can be of $rank(A) < N$, and therefore (non-pseudo) inversion methods are not applicable.

**Table 7:** Computation time for various fitting methods on a data set containing 800 mixtures, 256 Gaussians each. `torch.linalg.pinv` and `torch.linalg.lstsq` were tested, both functions being able to solve our problem. However, `lstsq` does not compute a gradient in the current pytorch version and would be therefore unusable. But even if so, the timings are prohibitive. Our heuristic takes only 3.8 milliseconds.

| $D =$ | $B$ | $concat(B, rand(N, k))$ | $rand(4N, k)$ |
|---|---|---|---|
| `torch.linalg.pinv` | 10.2s | 10.9s | 11.3s |
| `torch.linalg.lstsq` | 1.1s | 1.5s | 2.0s |

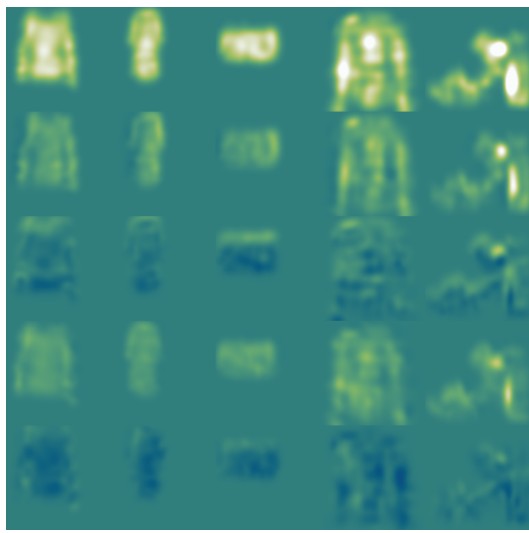

*Input*

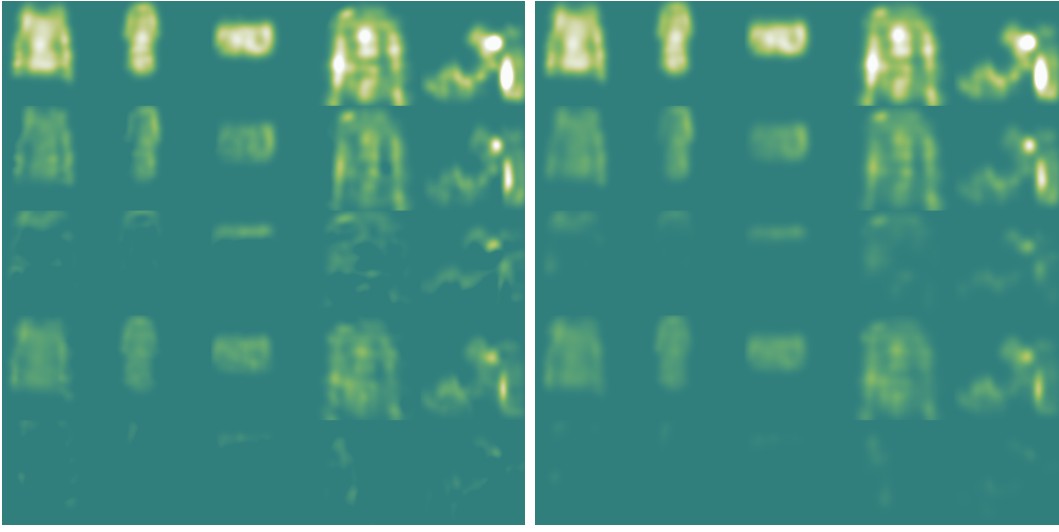

*Target ($\varphi$(input GM))*             *Heuristic*

**Figure 7:** Selection of the 800 Gaussian mixtures used for the evaluation of the ReLU fitting method.

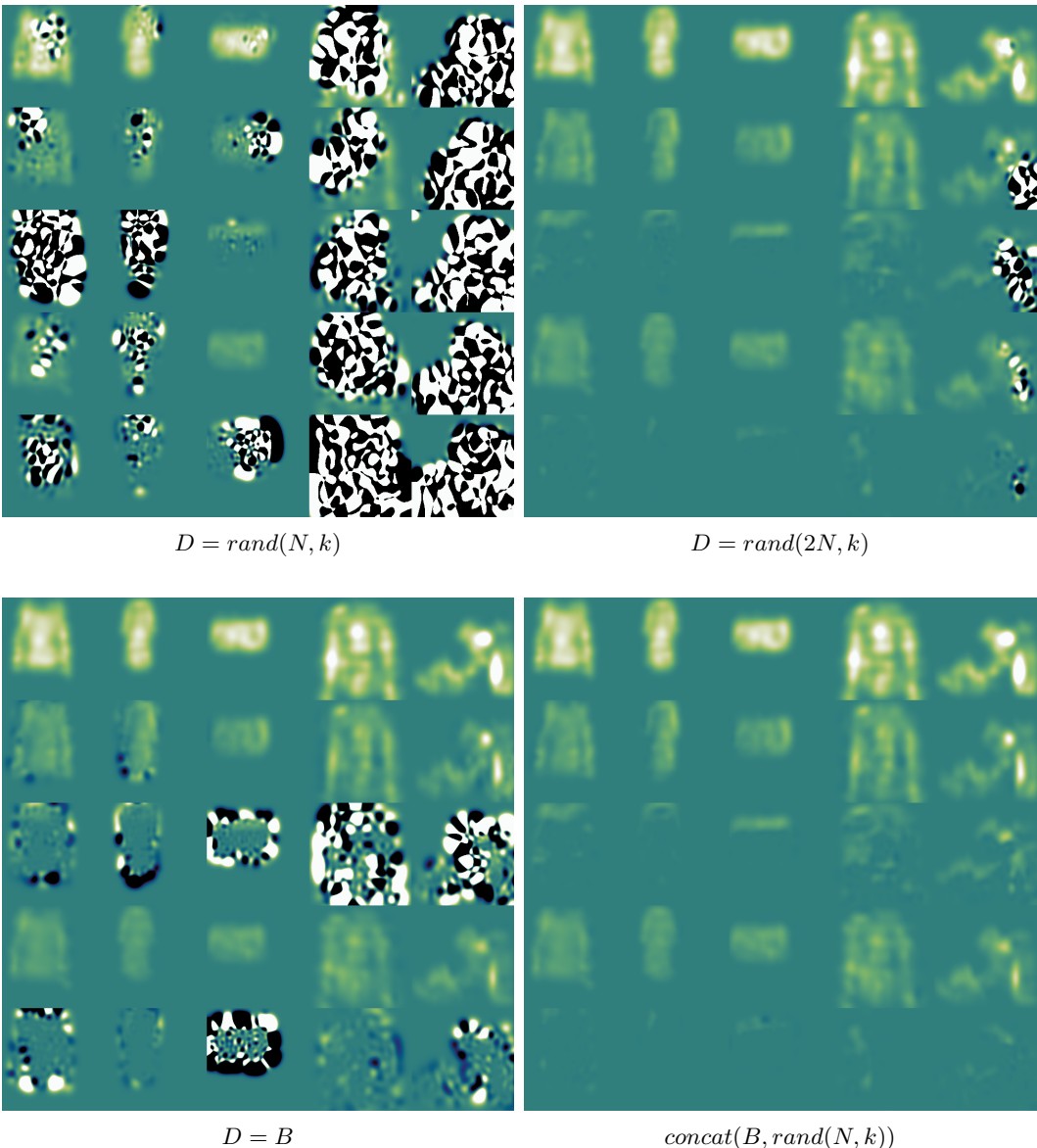

$$D = rand(N, k)$$

$$D = rand(2N, k)$$

$$D = B$$

$$concat(B, rand(N, k))$$

**Figure 8:** GM fitting using a least squares algorithm. We verified that the mixture fits the target at the sample positions, so the solution of the least-squares problem is correct in all cases. The mixture overshoots however and does not fit the ReLU well away from the sampling positions. Increasing the sample count, or selecting better samples helps, but even then there is no guarantee, that the mixture won't overshoot. `torch.linalg.lstsq` was used, but the results for `torch.linalg.pinv` are similar.

## C    DETAILS ON THE DENSE FITTING

The purpose of the dense fitting step in our approach is to fit a transfer function, for example, a ReLU. Our fitting method only computes new weights for the Gaussian mixture. Position and covariance matrices of the Gaussians are not changed (as described in Section 4.1). Accordingly, we give only equations for the new weights $a''$, which are computed in 2 steps.

**Coarse fitting.**    The first step computes a coarse, or initial approximation to the target ($\varphi(\mathrm{gm})$):

$$a'_i = \begin{cases} a_i, & \text{if } a_i > 0 \\ \epsilon, & \text{otherwise} \end{cases}, \tag{13}$$

where $a_i$ are the weights of the old mixture, and $i$ runs through all Gaussians. This design is motivated by two key considerations.

First, we argue that it is safer if the weights in the fitting are strictly positive. In our design, this will be the case, since $a' > 0$ and the second step does not introduce a negative sign. Negative weights themselves are not a problem for the GMCN architecture. However, the ReLU defines a function that is positive throughout the domain, and so should the fitting. This in itself still does not conflict with negative weights as long as the sum is greater than zero for all positions in the domain. However, our approach only considers the mixture's evaluation at the Gaussians' centroids, and negative weights could cause the sum to become negative at locations in-between these evaluated points. Something similar happens when using a linear solver (Section B). As a positive side effect, reduction fitting becomes simpler if it does not have to deal with negative weights.

Secondly, we scale the Gaussians (change their weight) as little as possible, therefore we do not initialize $a'_i$ simply to 1. The rationale here is, that we want to retain the shape of the mixture as much as possible without having to fetch the positions and covariance matrices from memory. Now, if we initialized the weights to 1, the result at the centroid would be approximately correct, but the shape would change more than necessary.

**Correction**    The new weights $a''$ are computed using a correction based on the transfer function

$$a''_i = a'_i \times \frac{\varphi\left(\mathrm{gm}(\boldsymbol{B}_i, \boldsymbol{a}, \boldsymbol{B}, \mathbf{C})\right)}{\mathrm{gm}(\boldsymbol{B}_i, \boldsymbol{a}', \boldsymbol{B}, \mathbf{C})}, \tag{14}$$

where $B$ contains the positions, $\boldsymbol{C}$ the covariance matrices, and $\varphi$ represents the ReLU. The fraction computes a correction factor for each Gaussian by evaluating the target function and the current approximation at the Gaussians centroid. This computation is $O(N^2)$ in the number of Gaussian components in terms of runtime. In order to avoid storage of intermediate results and a large gradient tensor in PyTorch, we implemented the evaluation in a custom CUDA kernel.

On the CPU it would be possible to trade accuracy for performance by ignoring contributions that are below a certain threshold, and using a tree-based acceleration structure (e.g., a bounding volume hierarchy). However, our experiments have shown, that this does not scale well on the GPU. The evaluation kernel is reasonably optimized. Still, due to a large number of Gaussians, which can easily go into several thousands, times the number of feature channels, times the batch size, these two evaluations account for roughly half of the overall training/test runtime.

In order to improve performance decisively, either the number of Gaussians would have to be reduced before this fitting step (e.g., during the convolution as outlined in our future work section), or a different algorithm could be used, or both.

## D    DETAILS ON THE REDUCTION STEP

The purpose of this step in our approach is to reduce the amount of data by fitting the *target* with a smaller mixture. It is applied after ReLU fitting, hence all weights are positive, and the mixture can be turned into a probability distribution by dividing the weights by their sum. Therefore, probability-based methods can be applied. In particular, it would be possible to sample the mixture and then use an expectation-maximization (EM) algorithm. However, that would be prohibitively expensive, given the potentially large number of Gaussians.

Since TreeHEM uses equations from the hierarchical EM algorithm by Vasconcelos & Lippman (1999), we start our detailed description with HEM and then continue with details on TreeHEM.

**Hierarchical EM**    Hierarchical EM starts with one Gaussian per point, which is the first level. In each following level ($l + 1$), a smaller mixture is fitted to the previous level ($l$). The new mixture needs to be initialized first, for instance by taking a random subset of the mixture in level $l$. In the E step, responsibilities are computed using

$$r_{is} = \frac{\mathcal{L}(\Theta_s^{(l+1)}|\Theta_i^{(l)})w_s}{\sum_{s'} \mathcal{L}(\Theta_{s'}^{(l+1)}|\Theta_i^{(l)})w_{s'}},$$

$$\mathcal{L}(\Theta_s^{(l+1)}|\Theta_i^{(l)}) = \left[g(\mu_i^{(l)}|\Theta_s^{(l+1)})e^{-\frac{1}{2}tr([\Sigma_s^{(l+1)}]^{-1}\Sigma_i^{(l)})}\right]^{\hat{w}_i}, \tag{15}$$

where $\hat{w}_i$ is the number of *virtual* samples (equations taken from Preiner et al. (2014)). In the original work, $N$ points were fitted, so $\hat{w}_i = Nw_i$ corresponded to the number of points that Gaussian was representing. Since we do not process points but Gaussians directly, $N$ becomes an implementation defined constant. Afterward, the M step can be taken:

$$w_s^{(l+1)} = \sum_i r_{is}w_i^{(l)}$$

$$w_{is} = r_{is}w_i/w_s^{(l+1)}$$

$$\mu_s^{(l+1)} = \sum_i w_{is}\mu_i^{(l)}$$

$$\Sigma_s^{(l+1)} = \sum_i w_{is}\left(\Sigma_i^{(l)} + (\mu_i^{(l)} - \mu_s^{(l+1)})(\mu_i^{(l)} - \mu_s^{(l+1)})^T\right), \tag{16}$$

where $w$, $\mu$, and $\Sigma$ are the weight, centroid, and covariance matrix, respectively, $i$ is the index of the (larger) target mixture, $s$ the index of the new mixture. In each level, only one iteration is taken (E and M step), and the algorithm continues with the next level. Preiner et al. (2014) propose a geometrical regularization, in which $r_{is}$ is set to zero if the Kullback-Leibler divergence between Gaussians $i$ and $s$ is larger than a certain threshold.

We were not able to use hierarchical EM due to multiple reasons, including difficult parallelization on the GPU, low test accuracy on our data, poor performance (since every Gaussian needs to be tested against each other Gaussian), and the memory footprint of the corresponding gradient.

**TreeHEM**    We, therefore, developed the TreeHEM algorithm to improve on these issues. In short, we build a spatial tree of the (larger) target mixture and use it to localize the fitting, i.e., limit the fitting to a fixed-size subset of Gaussians in proximity of each other.

The tree is built using the same algorithm, as is described by Lauterbach et al. (2009) for building bounding volume hierarchies of triangles (LBVH), without the bounding boxes. In our case, we take the Gaussians' centroids, compute their Morton code, and use it to sort the mixture. This puts the Gaussians on a Z-order curve. It is then possible to build a tree in parallel by looking at the bit pattern of the sorted Morton codes. Please consider the excellent paper from Lauterbach et al. for more details. Storing the indices of children and parents in the tree allows us to traverse bottom-up and top-down.

The tree is traversed twice. In the first, bottom-up, pass we collect and fit Gaussians, and then store them in a cache with up to $T$ Gaussians in the tree nodes. Bottom-up traversal is implemented in a parallel way. Threads in a thread pool start from leaf nodes. The first thread reaching an inner node simply sets a flag and is re-entered into the pool. The second thread to reach an inner node does the actual work: First, the Gaussians from the caches of the two children are collected, resulting in up to $2T$ Gaussians. If the children are leaves, or not yet full, then that number can be below $T$, in which case they are written into the cache. Otherwise, $T$ new Gaussians are fitted to the collected ones, stored in the cache, and traversal continues until reaching the root node.

For the fitting, we first select $T$ initial Gaussians from the $2T$ collected ones. This is done by dividing the $2T$ Gaussians into $T$ clusters based on the Euclidean distance between the Gaussians' centroids. From each cluster, the Gaussian with the largest weight is selected, giving $T$ initial Gaussians. The

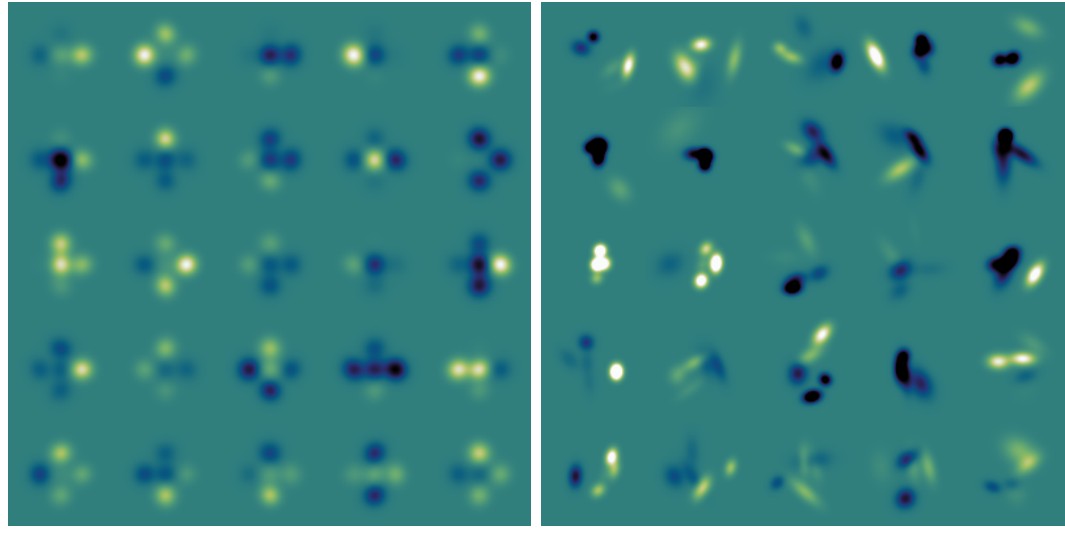

*Initialization*                                      *After about 1 million training steps*

**Figure 9:** Convolution kernels of a 2D data set. It is clearly visible, that all of the parameters (weights, positions, and covariance matrices) adapt to the data, once training progresses.

rationale behind this method is to try to select the most important Gaussians that have the least overlap. We then use Equations 15 for the E step, and Equation 16 for the M step of the fitting, where we set $l + 1$ to the initial mixture described above, and $l$ to the target mixture consisting of $2T$ Gaussians.

At this point, we obtain a tree with a mixture of $T$ Gaussians in each node that fits the data underneath. In the second, top-down, pass we aim to select $N/T$ nodes that best describe the target mixture, where $N$ is the number of Gaussians in the fitting. This was implemented using a greedy algorithm with a single thread per mixture (which is not a problem as the kernel only traverses the very top part of the tree). All selected nodes are stored in a vector, where the first selected node is the root. In each iteration, we search through the vector and select the node with the largest mass (integral of Gaussians, i.e., sum of weights) and replace it with its two children. This adds $T$ nodes to our selection in each iteration. The iteration terminates once we selected $N/T$ nodes. Finally, we iterate through these nodes and copy the Gaussians from the cache into the destination tensor.

**Gradient computation**    Since all of the above is implemented in a custom CUDA kernel, we do not have the usual automatic gradient computation. Therefore we had to compute the gradient manually. To this end, we implemented unit tests of our manual gradient, which compare our manual gradient against a gradient computed by the autodiff C++ library (Leal, 2018).

## E    KERNEL EXAMPLES

Figure 9 shows examples of training kernels for a 2D data set.

## F    FURTHER RESULTS AND ABLATION

In this section, we include the full suite of results collected on the ModelNet10 and ModelNet40 data sets. Unless stated otherwise, the following tables report the test accuracy as shown in TensorBoard with the smoothing set to 0.9. Times shown are for training on an NVIDIA GeForce 2080Ti, without the fitting of the input data. As part of the ablation, we consider different network configurations and lengths. The network notation indicates Gaussian convolution layers (convolution, activation, and pooling) using arrows ($\rightarrow$) and specifies the data dimensions between these layers by the *number of feature channels / number of data Gaussians*.

**Table 8:** ModelNet10 with TreeHem($T = 2$) fitting and batch size 14, evaluating varying network lengths.

| Model Layout | Epochs / acc. | | #kernels | #params | mem | tr. time (60) |
|---|---|---|---|---|---|---|
| | 50 | 60 | | | | |
| 1/128 → 10 | 74.73 | 72.45 | 10 | 651 | 0.3GB | 0h 36m |
| 1/128 → 8/64 → 10 | 89.37 | 89.36 | 88 | 5721 | 0.3GB | 2h 6m |
| 1/128 → ... → 16/32 → 10 | 91.06 | 91.17 | 296 | 19241 | 0.7GB | 4h 49m |
| 1/128 → ... → 32/16 → 10 | 91.83 | 91.99 | 968 | 62921 | 1.6GB | 10h 10m |
| 1/128 → ... → 64/8 → 10 | 92.44 | 92.61 | 3336 | 216841 | 3.4GB | 21h 22m |
| 1/128 → ... → 128/4 → 10 | 92.22 | 92.30 | 12168 | 790921 | 5.9GB | 41h 22m |

**Table 9:** ModelNet10 with TreeHem($T = 2$) fitting and batch size 21, testing number of input Gaussians. For all layouts, the number of kernels is 3,336, and the number of parameters is 216,841.

| Model Layout | Accuracy |
|---|---|
| 1/128 → 8/64 → 16/32 → 32/16 → 64/8 → 10 | 92.78 |
| 1/64 → 8/64 → 16/32 → 32/16 → 64/8 → 10 | 92.28 |
| 1/32 → 8/64 → 16/32 → 32/16 → 64/8 → 10 | 91.44 |
| 1/32 → 8/32 → 16/32 → 32/16 → 64/8 → 10 | 92.09 |
| 1/16 → 8/64 → 16/32 → 32/16 → 64/8 → 10 | 90.92 |
| 1/16 → 8/16 → 16/16 → 32/16 → 64/8 → 10 | 89.91 |
| 1/16 → 8/8 → 16/4 → 32/2 → 64/2 → 10 | 90.17 |

## F.1 MODELNET10

In Table 8, we report the classification accuracy of different GMCN models that was achieved after 50/60 epochs, respectively, as well as their resource consumption. The simplest tested design uses a single feature channel and 128 fitted Gaussians for the input layer, followed by a 10-way classification. With the addition of convolution layers, the classification performance improves. The first internal convolution layer uses 8 input feature channels and 64 fitted Gaussians, which suffices to significantly raise accuracy. To test longer networks, each row in the table builds on the last and inserts an additional convolution layer just before the final classification layer. With each new layer, the number of feature channels is doubled and the number of fitted Gaussians halved. The results demonstrate the GMCNs of variable length exhibit consistent behavior. After 50 training epochs, the classification accuracy no longer changes drastically and usually increases slightly given another 10 epochs. Classification accuracy eventually peaks using four internal convolutional layers and slightly drops when a fifth layer is added.

In Table 9, we investigate the effect of varying the number of fitted Gaussians in each layer on classification accuracy. Starting from the best-performing model identified previously with four internal convolution layers, we assess how adapting the internal fitting can influence classification accuracy. Naturally, the network is sensitive to changing the amount of information it receives, i.e., the number of Gaussians fitted to the input data. However, the same is not generally true for the internal layers. The GMCN architecture appears to favor distinct numerical relations between fitted Gaussians in the individual layers. For example, quartering the number of input Gaussians (32 vs. 128) without adapting the remainder of the network yields a poorer performance than using only half as many fitted Gaussians in the first convolution layer (32 vs. 64).

In Table 10, we compare the different available versions for expectation maximization (EM) in the fitting of Gaussians: modified EM (MEM), and two variations of TreeHEM, with parameters T = 2 and T = 4, respectively. These results are obtained using data sets with 64 fitted input Gaussians. We examine the error of the two fitting processes (dense fitting of the transfer function, and reduction / pooling) in three measurements. Measurement is done by computing the RMSE on evaluations from several 100k sampled positions of the fitted GM and the input to the respective fitting process. Our 3 measurements are:

**Table 10:** Comparing different fitting methods in a *1/64 → 8/32 → 16/16 → 32/8 → 10* GMCN with batch size 21 on ModelNet10. We report the RMSE of each fitting against the ground truth in each layer.

| Layer | Method | RMSE | | |
|---|---|---|---|---|
| | | **all** | **ReLU** | **reduction** |
| Convolution Layer 1 → 8 | MEM | 0.4457 | 0.1639 | 0.2653 |
| | TreeHEM (T=2) | 0.359 | 0.2253 | 0.1144 |
| | TreeHEM (T=4) | 0.3487 | 0.2204 | 0.1159 |
| Convolution Layer 8 → 16 | MEM | 0.4991 | 0.1433 | 0.2694 |
| | TreeHEM (T=2) | 0.208 | 0.1031 | 0.0717 |
| | TreeHEM (T=4) | 0.223 | 0.1104 | 0.0806 |
| Convolution Layer 16 → 32 | MEM | 0.3582 | 0.1792 | 0.0842 |
| | TreeHEM (T=2) | 0.2206 | 0.1346 | 0.0469 |
| | TreeHEM (T=4) | 0.2461 | 0.1259 | 0.0707 |

- *all*: overall fitting error, compares the output of the reduction step with ground truth transfer function evaluations of the input.

- *ReLU*: compares the output of the dense fitting step with ground truth transfer function evaluations of the input.

- *reduction*: compares the output of the reduction step with the input to the reduction step (ouput of the dense fitting).

Comparing the two, we find that the ReLU fitting error is significantly larger in each layer. This suggests that GMCN performance is affected more by the fitting accuracy than by the reduction of Gaussians. Note that in general, **all ≠ ReLU + reduction**.

Table 11 confirms this trend when using input data sets with 128 fitted input Gaussians. However, it also illustrates our preference for using the TreeHem fitting algorithm using T = 2. The MEM method runs out of memory with this configuration, and therefore the errors cannot be evaluated. Comparing TreeHEM with T = 2 and T = 4, we find that the first variant, in addition to reduced resource requirements, yields both a lower fitting error and a better accuracy. As stated before, we attribute this behavior to the smearing of Gaussians when a larger parameter T is used (more Gaussians are merged simultaneously and thus fine details can be lost).

Table 12 illustrates the development of the test accuracy with the number of epochs allowed for fitting Gaussians to the 3D input data set. These experiments mainly serve to identify the suitable number of fitting iterations to be used in our evaluation. A baseline of 80% is quickly achieved with all approaches after only 5 epochs. Between 50 and 60 epochs, there is little discernible improvement of the accuracy. In terms of the fitting techniques to use, we find that in all cases, k-means performed best for the ModelNet10 data set, further solidifying our previous results.

## F.2 MODELNET40

In order to assess GMCNs in the context of a problem with higher complexity, we ran a series of tests for the ModelNet40 data set. Due to its larger size and the fact that our proof-of-concept GMCN implementation is not heavily optimized, the number of experiments and amount of hyperparameter exploration that could be performed in a timely manner was limited. Data augmentation was not possible, because that would multiply the number of training samples and therefore runtime. Out of the tested variants, GMCNs performed best with a model that contains four internal Gaussian convolution layers, similar to our findings for ModelNet10. The particular configuration we used is similar, with the exception that the last layer performs a 40-way classification (*1/128 → 8/64 → 16/32 → 32/16 → 64/8 → 40*). Table 13 lists the achieved test accuracy with different batch sizes and epochs. This data confirms that GMCNs converged: going from 50 to 60 epochs only has a small effect on accuracy, improving it slightly. The best performance was achieved with a test accuracy of 87.6%, using a batch size of 14, training for 60 epochs.

**Table 11:** Comparing different fitting methods in a *1/128 → 8/64 → 16/32 → 32/16 → 64/8 → 10* GMCN with batch size 21 on ModelNet10. We report the RMSE of each fitting against the ground truth in each layer.

| | | RMSE | | |
|---|---|---|---|---|
| **Layer** | **Method** | **all** | **ReLU** | **reduction** |
| | MEM | out of memory | | |
| Convolution Layer 1 → 8 | TreeHEM (T=2) | 0.3391 | 0.2177 | 0.0948 |
| | TreeHEM (T=4) | 0.3477 | 0.228 | 0.1011 |
| | MEM | out of memory | | |
| Convolution Layer 8 → 16 | TreeHEM (T=2) | 0.2116 | 0.1121 | 0.0688 |
| | TreeHEM (T=4) | 0.1859 | 0.1032 | 0.065 |
| | MEM | out of memory | | |
| Convolution Layer 16 → 32 | TreeHEM (T=2) | 0.2085 | 0.1415 | 0.0334 |
| | TreeHEM (T=4) | 0.236 | 0.1524 | 0.0468 |
| | MEM | out of memory | | |
| Convolution Layer 32 → 64 | TreeHEM (T=2) | 0.1204 | 0.0724 | 0.026 |
| | TreeHEM (T=4) | 0.156 | 0.0764 | 0.045 |

**Table 12:** Comparing different input fitting methods in a *1/128 → 8/64 → 16/32 → 32/16 → 64/8 → 10* GMCN with batch size 21 on ModelNet10.

| | Epochs / acc. | | |
|---|---|---|---|
| | **5** | **50** | **60** |
| k-means | 84,06 | 93,11 | 93,29 |
| k-means+EM | 82,36 | 92,72 | 92,81 |
| random+EM | 80,42 | 91,94 | 92,13 |
| Eckart | 80,48 | 90,82 | 91,04 |

**Table 13:** Different batch sizes and training epochs on our best-performing GMCN with four internal convolution layers (1/128 → 8/64 → 16/32 → 32/16 → 64/8 → 40) for ModelNet40.

| | Epochs / acc. | | | | |
|---|---|---|---|---|---|
| **batch size** | **50** | **60** | **#kernels** | **#params** | **training time (60)** |
| 14 | 87,35 | 87,62 | 5256 | 341641 | 2d 14h 57m |
| 21 | 86,65 | 87,01 | 5256 | 341641 | 2d 12h 7m |

**Table 14:** Results, parameters and training time for training and testing a GMCN model with a total of five convolution layers (1/128 → 8/64 → 16/32 → 32/16 → 64/8 → 128/4 → 40) on ModelNet40.

| | Epochs / acc. | | | | |
|---|---|---|---|---|---|
| **batch size** | **50** | **60** | **#kernels** | **#params** | **training time (60)** |
| 14 | 77,84 | 77,82 | 16008 | 1040521 | 4d 23h 17m |

In contrast to our results for ModelNet10, the achieved accuracy trails behind the best-performing state-of-the-art approaches when applied to ModelNet40. It must be noted that these competitors include view-based methods and architectures that rely on purpose-built feature learning modules (e.g., relation-shape in RS-CNN or reflection-convolution-concatenation in NormalNet RCC). Hence, our results are in fact competitive with other established convolution-based baselines, such as VoxNet (83%), O-CNN (90%), and NormalNet Vanilla (87.7%).

However, there are several options for further increasing the performance of GMCNs that were not explored in this work. For example, due to the long training times of our proof-of-concept implementation, we did not use data augmentation in the training process, making the model more susceptible to overfitting. Indeed, the training accuracy peaked at 95%, which is 8% higher than the test accuracy. Thus, a non-negligible amount of overfitting in our best-performing model for ModelNet40 seems likely. Lastly, we expect that one important factor for maximizing GMCN performance is the precision of the fitting of activation functions, as outlined below.

Similar to ModelNet10, the network's accuracy for ModelNet40 drops when increasing the number of convolution layers to a total of five (see Table 14). Rather than overfitting, we ascribe this trend to the imprecise fitting of activation functions. This is supported by the results reported in Tables 10 and 11, which attribute the majority of the fitting error to ReLU fitting. Hence, we expect that the accuracy of GMCNs can be improved with more elaborate, high-quality fitting methods, which we hope to explore in future work.

## G  Activation in parameter space (applying the transfer function to the weights)

At first glance, it seems possible that ReLU fitting in GMCNs could be simplified by applying the transfer function directly in parameter space, i.e., on the weight vector. In other words, performing an approximation of Gaussians by Diracs for the purpose of activation to reduce the complexity of the approach.

Unfortunately, that is not the case. Currently, if a negative Gaussian is on top of a positive one, then the negative would reduce the contribution of the positive one. In extreme cases, the positive Gaussian could even disappear entirely. This would not be possible if the ReLu would be applied directly to the weight: All negative Gaussians would be cut away, while the positive ones would simply pass through. Thus, no gradient would ever reach a negative kernel Gaussian. Consequently, positive kernel Gaussians, in conjunction with the convolution and such a parameter space activation, would act as if there were no activation at all.

