# OpenReview forum: "Gaussian Mixture Convolution Networks"
_ICLR.cc/2022/Conference — ICLR 2022 Poster_

### Official Review · Reviewer_RmKw · 2021-10-28

**Correctness:** 3
**Technical Novelty And Significance:** 3
**Empirical Novelty And Significance:** 3
**Recommendation:** 6
**Confidence:** 4

**Details Of Ethics Concerns:**


N. a.

**Main Review:**

Strengths

This is an interesting work, where the main goal is to replace conventional/standard CNN operations by GM. And this, of course, can  open the door to other innovative approaches that substitute the conventional ones.

Weaknesses

Although I find the proposed approach interesting I would expect some more experimental evaluation to other related approaches and other datasets. Nevertheless, this does not substantially hamper the paper. Another issue that I was struggling with, and I think the main bottleneck of the paper, is the reduction step (namely the use of M Gaussians). And perhaps other choices for eq. (6) and (7) that are not discussed (is it possible to have alternatives ?). See my concern in the detailed review.



**Summary Of The Paper:**

CNNs have made an undeniably successful/impact on computer vision tasks. Its core operation relies on grids of discrete data samples, where structure representation of data is used (e.g. vector matrices …). This paper tries to disrupt this well established idea used so far, by proposing a Gaussian of mixtures (GMs) for tackling the curse of dimensionality, that is not avoidable with the conventional CNN architectures. The resulting architecture is termed as a deep functional network.

**Summary Of The Review:**


When describing the TreeHEM approach, for reducing/merging  (or killing some Gaussian components) I found it rather cumbersome as other methodologies can be used to prune the mixture components. Next, I leave several questions for the discussion:

-it is known that the EM suffers from the problem of the initialization, that is, how many components (per node)  should we represent the data. This often leads to a model selection type of problem. So, how can we deal with this issue automatically in the proposed architecture ?

-Also, it is mentioned that reduction of  the number to Gaussians must be performed. Here, it would be welcome to introduce some of the statistical metrics that are also available to perform this task. In this way it could be possible to have more than one approach, from which a comparison could be performed, and thus, providing more insight into how this could be properly tackled.

-It is not clear the mapping between the proposed context and the clustering hierarchy in Vasconcelos & Lippman (1999) work (that is, levels, blocks etc…).

It is mentioned that a value of T = 2 Gaussian is the best choice. However, this is difficult to follow …

-For completeness, please specify the meaning of the operation in (8).

-When performing the pooling, the fitting process is also performed.Here, it is necessary to enlarge the receptive field. To accomplish this,  the reduction of the covariance diagonal must be decreased. The question is, how is it possible to find the best scaling factor for this purpose ? receptive field . Is it possible to act on the covariance uncertainty for this purpose ?

-it is said that the k-means is used to estimate the initial center’s position. What happens if this procedure does not provide the best location for the centroids ?It is well known that this algorithm may have some limitations regarding this issue.

-Since the proposal is based on the EM algorithm, a clear mapping with this should be done. Specifically, to describe explicitly the E-step and M-step

---

> ### Author Response · Authors · 2021-11-20
> **Answer to Reviewer RmKw**
>
> Thank you for taking the time to review our paper and provide your comments! We will try to address any open questions below.
>
> >Possible extension of the evaluation.
>
> We will gladly add more data points on ModelNet40 and an analysis thereof (please see our answer to Ao7K for full details due to length limitations).
>
> >Heuristic using Equations 6 and 7.
>
> We are uncertain about how to interpret the question. Is it whether there are alternatives to the heuristic? We mention in the paper that a solver-based approach is infeasible due to computational requirements. We consider the problem of transfer function fitting a difficult one and are constantly looking for better options. One alternative would be to sample the mixture and use an EM fitting approach. This is currently not feasible due to the large number of Gaussians but may become realistic if the reduction step is integrated with the convolution operation. We will add corresponding notes when describing the heuristic.
>
> >TreeHEM for reduction / merging or pruning of Gaussians.
>
> TreeHEM may appear cumbersome at first, but it is a practical necessity to improve performance over a more straightforward, EM-based algorithm due to time and memory constraints, which we show in Section 5.3 and Table 3. One alternative would be to employ a method such as Preiner et al. (2014), which we cite in our paper. However, among other issues, this does not parallelize well. We also easily. We do not expect that actual pruning, i.e., removing part of the Gaussians, would work well. Typically, the convolution outputs thousands of Gaussians, and these are then reduced by fitting to, e.g., 64. The gradient would not be able to flow sufficiently. We did test pruning after fitting as a regularisation technique similar to dropouts, but it did not improve the results.
>
> >EM initialisation and statistical metrics of the fitting:
>
> We are looking forward to discussing these topics with the reviewer, but to give a clear answer, we would require additional clarification. With regard to EM initialisation, we are unsure which portion of the paper the question is referring to. Is it regarding the creation of the Gaussian mixtures for the input data sets? Or is it regarding the mixture-to-mixture EM fitting? Or its replacement, the TreeHEM algorithm, i.e., how the number of components in each tree node is chosen?
>
> A similar issue came up concerning the suggested use of statistical metrics for the reduction of Gaussians. Does this imply that we should provide statistical metrics to quantify the quality of our fitting to enable future comparison with other reduction techniques? Or does it suggest that the reduction itself might be replaced by a method that is based on statistics? If so, would you kindly provide an example of such a technique so we can begin our discussion with a pertinent comparison?
>
> >Vasconcelos & Lippman and TreeHEM:
>
> We note that we do not use Vasconcelos & Lippman's algorithm, only their equations. Hence, there exists no direct mapping between their hierarchy and ours. Their sample block corresponds to a Gaussian, though we do not start with samples in the first place. Hence such a mapping is questionable. Their first level would correspond to our tree leaves, but during bottom-up merging, Gaussians from different levels are merged since the tree is not necessarily balanced. Space permitting, we will include corresponding remarks in the revision.
>
> >Choice of T
>
> According to our experiments, T = 2 is faster than T = 4, while keeping the fitting error almost the same or better, as we showed in the results section (5.3, and Table 3). In accordance with other reviewers’ requests, we will be adding an interpretation for T’s effects on runtime (linear in memory/performance) and accuracy (smearing of Gaussians and loss of fine details during reduction).
>
> >Meaning of the operation in Eq. 8:
>
> The explicit definition (i.e., the Hadamard product) was lost when preparing the paper for submission; thanks, we will fix it in the revision.
>
> >Pooling operation and scaling:
>
> We had difficulties interpreting this question, particularly the mention of covariance uncertainty. Does this refer to the off-diagonal entries of the covariance matrix? To what end / effect should the reduction of the diagonal be decreased? Interpreting this phrase correctly is tricky for us since "reduction" in our submission can refer to multiple concepts. We look forward to the reviewer’s response and a productive discussion!
>
> >K-means initialisation of the input fitting:
>
> Our method is robust with regard to such suboptimal centroids, as that Gaussian would have a small weight, and others would be used to represent the data. In our experience, GMCNs can deal with this reasonably well, even if some Gaussians are missing completely.
>
> >Clear mapping between EM and our algorithms:
>
> We will provide a more formal mapping to EM, that is, a distinction of explicit expectation and maximization step. Thanks for this hint

---

> ### Comment · Reviewer_RmKw · 2021-11-29
> **Finishing the review**
>
>
> I have read all the comments formulated. Even though some issues of mine were not answered, others were clarified which allowed me to clarify some other points that were not raised before. This is to say that I maintain my initial score given.

---

> ### Comment · Reviewer_eQt8 · 2021-11-29
> **Finishing the review**
>
> Having read all reviews comments and answers, I still think the paper is worth publication for its novelty of flowing functions (i.e. gmm) through a conv net. The proposed non linear step may not be great (i.e. not mathematically elegant, not computationally efficient, etc.),
> and I think improving on it is an interesting future research question for the ML community  for subsequent works.

---

### Official Review · Reviewer_eQt8 · 2021-10-30

**Correctness:** 4
**Technical Novelty And Significance:** 4
**Empirical Novelty And Significance:** 2
**Recommendation:** 8
**Confidence:** 4

**Main Review:**

 In a similar spirit to  Functional Data Analysis (Ramsay& Silverman) (e.g. with Functional PCA), the paper proposes to learn  functions with neural networks
which is appropriate when data can be well embedded into GMs (e.g. 3d point cloud).
The paper proposes approaches for  GM fitting so that the output of the activation function (ReLu) becomes a GM.
 As the convolution of GMs produces a GMs with more components (Gaussians), reduction techniques are used to limit the number of components in the GMs as these pass through the network.
The resulting proposed approach shows good performance for classification.

 The cross product (convolution) of two GMMs has a closed form solution. (e.g. as already exploited  for   registration  https://doi.org/10.1109/TPAMI.2010.223 and in statistics http://www.jstor.org/stable/1271214 ).
Equations  could be better presented e.g. in Eq (1) the weight $a$ should not be part of the Gaussian (as it integrates to  1) and Eq. 2 would best be written as a weighted sum. Results Eq (3) and (4) are known so section 3 acts as a reminder/ state of the art.
Training time are said to not be competitive against state of the art.

Given that  GMs are chosen as functional representation,  why is ReLu chosen as activation? is there any other activation functions that would have transformed a GM into a GM (apart from the identity function) ?
For simplicity why is the activation function (ReLu) not applied to the means only (i.e. approximating Gaussian by Dirac ) while applying identity to covariances? (i.e. why is activation not applied in the parameter space of the Gaussian ?)


**Summary Of The Paper:**

The paper proposes to rethink deep learning for learning functions
(with functions being gaussian mixtures GMs, with  positive and/or negative weights unlike for PDFs) as opposed to tensors traditionally used for embedding data.
Filters are learned as GMs since the cross product (convolution) of two GMMs has a closed form solution.
Applying  an activation/transfer function (e.g. ReLu) to a GM does not produce a GM and
the paper proposes to approximate a GM from the output of the activation function. Such approach allows to define layers with input/output in GM forms.


**Summary Of The Review:**

The paper is interesting and novel as it rethink neural networks  for learning functions.

---

> ### Author Response · Authors · 2021-11-20
> **Answer to Reviewer eQt8**
>
> Thank you for the time and effort put into this review; we will try to address all raised questions below.
>
> >Style of Section 3
>
> We agree, Section 3 acts mostly as a reminder, but we think that this is warranted since we do not expect all targeted readers to know these facts by heart. Regarding the change of notation in Equations 1--4, we will happily modify them to match the suggested style.
>
> ---
>
> >Training time not competitive
>
> The training time with the presented implementation trails behind state-of-the-art. However, we would like to highlight that our conclusion includes suggestions that enable us to significantly alleviate these problems. The paper focuses on our main contribution, the GMCN architecture itself, and leaves a detailed exploration of these optimizations to future work.
>
> ---
>
> >Why ReLU is chosen as activation function?
>
> We would like to clarify that GMCNs allow a range of possible activation functions. Any non-linear function that takes a Gaussian mixture as input and returns one could work. The presented fitting-based approach is only one example of this. If that approach is chosen, any conventional transfer function could be used, as long as a suitable fitting method is provided. Our focus on ReLU was mainly motivated by its predominant use in deep learning. In fact, it may be possible to build special transfer functions that are designed with effective fitting in mind, an avenue that could be pursued in future work.
> With regard to approximating Gaussians with Diracs for the ReLU fitting (i.e., activation in parameter space): Basically, this would imply applying the transfer function before summing the Gaussians. As a consequence, negative Gaussians wouldn't have a chance to influence the positive ones. However, one definitive contribution of our approach is its ability to also evaluate negative Gaussians as part of training and inference. We have received similar questions previously. Hence, to improve clarity, we will address this in an appendix, including a figure visualizing these relations.

---

### Official Review · Reviewer_gTB4 · 2021-11-03

**Correctness:** 3
**Technical Novelty And Significance:** 4
**Empirical Novelty And Significance:** Not applicable
**Recommendation:** 6
**Confidence:** 3

**Main Review:**

## Strengths
+ Novel, innovative approach
+ Tackles an important problem

## Weaknesses
- Somewhat unclear how small the memory footprint is compared to other methods
- Some central elements like dense fitting and reduction step not very well described
- Somewhat unclear what the limiting factors are for scaling to more complex datasets


## Detailed comments on weaknesses

### Memory footprint

A key argument for the method is that it allows working with sparse (e.g. volumetric) data without running into memory issues. However, after reading the paper I don't have a clear idea about the memory footprint. The number of features (8–>64) and the number of Gaussians (128–>8) doesn't seem very large, yet Table 3 reports ~6 GB memory (these numbers are not very useful, since we don't know the batch size). I wonder whether other methods like OctNets, PointNet etc. wouldn't perform better when matching the memory footprint instead of the number of parameters. After all, memory is the limiting factor when dealing with high-D data, not the number of parameters.


### Central elements not described well

While I understand the motivation and goals behind the dense fitting and reduction step, I could not follow the explanations in section 4.1. Since it's a fairly central part of the method, I think this section needs substantial improvement, both in terms of why things are done the way they are (e.g. Eq. 6+7) and how they are done exactly (TreeHEM).

*[Update on this point after discussions] After some clarifications on the method, I am somewhat unsure about the role of potential negative weights in the convolution kernels. It appears that the method effectively does not use them (or, if so, in a very indirect way).*


### What limits applicability to more complex datasets?

It seems like the methods doesn't work (yet) on ModelNet40. I wonder why this is the case. Is the number of Gaussians and/or feature maps that can be used too small? If that's the case, then does the argument about having a memory-efficient method really hold? If that's not the case, then what is the limiting factor?


**Summary Of The Paper:**

The paper proposes a new convolution method to process sparse data in a memory-efficient manner. The authors represent input data, intermediate feature maps and convolution filters as mixtures of Gaussians, which allows computing the result of the convolution analytically. Since the number of mixture components would increase quickly from layer to layer and application of ReLU destroys the property of intermediate layers to be mixtures of Gaussians, they re-fit intermediate feature maps and develop a heuristic to reduce the number of mixture components. The authors evaluate their proposed method on MNIST (2d) and ModelNet10 (3D) and show competitive performance compared to classic PointNet/++ methods.

**Summary Of The Review:**

I am a bit torn about the paper in its current form. On the one hand it proposes a new and potentially very useful approach. On the other hand, the execution leaves some things to be desired. This makes it a bit difficult to properly assess the benefits and limitations of the method compared to other approaches, in particular more recent ones than PointNet/++. If the authors can clarify the issues mentioned above and provide a convincing argument that the method might scale also to more complex use cases, I'm willing to increase my score.

### Update after rebuttal/discussions
I am still a bit torn, since the intuitions behind e.g. the dense fitting step are still not clear and the role of negative weights. Since some points were clarified, I would give the authors the benefit of the doubt, though.

---

> ### Author Response · Authors · 2021-11-20
> **Answer to Reviewer gTB4**
>
> Thank you for the very detailed and structured review. Please find our responses to your comments below.
>
> >Memory footprint
>
> In conventional CNNs, the convolution operation, that is “per-pixel kernel-sized window, element-wise multiplication, and sum”, are usually built into one compact function, or CUDA kernel. Accordingly, intermediate results are never stored in memory. The prototype described in this paper is not compact in the same way. Intermediate results are explicitly stored because the prototype was easier to implement and explain this way. This is costly in terms of memory and computation. The integration of a fitting step directly in the convolution is possible with our approach but more complex compared to conventional CNNs. Hence, we argue that the overall potential of our approach is best expressed via its theoretical minimal memory footprint. The theoretical minimum memory footprint of GMCNs is linear in the number of feature channels (C) and the resolution in each convolution layer after fitting (N), hence: O(C * N). Based on these figures, our approach can provide a clear benefit in terms of minimal required memory consumption over other grid-convolution-based 3D or higher-dimensional methods. We optimistically assumed that this relation was implicitly made clear to readers from our method’s description. However, upon revisiting the text, we agree with the reviewer that this must be stated more explicitly in the paper. To this end, we will add an appendix that derives the theoretical memory footprint for such a compact implementation. We will also adapt our main text to reference it and emphasize its importance for the potential of our approach.
>
> To answer the reviewer's question regarding batch size: our tests ran with it set to 21, which we will add to the paper. As mentioned above, achieving the outlined minimal memory footprint in practice requires multiple optimizations and specializations of our method. We felt that a detailed description of these would be outside the scope of this first paper on GMCNs (though we include a preview of the necessary steps in our conclusion). In order to prioritize explainability in the foundational aspects of GMCN, we chose to describe a proof-of-concept implementation, which does not implement these specializations. Corresponding statements regarding the focus of the presented implementation will be added to the paper.
>
> Nonetheless, our basic implementation shows roughly comparable memory requirements as the mentioned convolution-based competitors, OctCNN and O-CNN. For lack of explicit figures, we conclude from their description that PointNet very likely has lower memory requirements; however, it is not a convolutional architecture. To improve our evaluation, we will add these numbers and remarks to our paper, as suggested by the reviewer.
>
> ---
>
> >Description of central elements
>
> We agree with the reviewer’s verdict that the concepts in Section 4.1 warrant a better description to improve clarity. Apart from rephrasing, we may not be able to expand these parts significantly in the main text due to paper length limitations, but we will include more detailed descriptions in the appendix and refer to them where appropriate.
>
> ---
>
> >Applicability to complex datasets
>
> The current limits in the applicability of the described implementation stem from its runtime performance. For instance, we did not use data augmentation for ModelNet40 due to the implied increased training time (compare reported training time in Section 5 without data augmentation). As noted in our discussion, implementing the convolution operation in a way that directly incorporates fitting can drastically improve performance and reduce memory footprint at the same time. To improve clarity, we will include these statements in our discussion.
>
> Furthermore, fitting accuracy in TreeHEM and ReLU activation is likely to have a significant influence as well. Table 3 relates the fitting accuracy of TreeHEM to test accuracy, and we will add a reference to it in our conclusion.
>
> Finally, we would like to qualify our current remarks in the paper regarding performance on ModelNet40 and FashionMNIST. While the achieved accuracy on these data sets is in fact on par with prior 3D convolution-based baselines (ModelNet40: ours 87.6%, VoxNet 83%, O-CNN 90%, NormalNet Vanilla 87.7%), they trail behind state-of-the-art results achieved with alternative, non-convolution-based approaches. These include view-based methods and architectures that rely on purpose-built feature learning modules (e.g., relation-shape in RS-CNN or reflection-convolution-concatenation in NormalNet RCC). The statement regarding the “simple GMCN architecture” is in reference to the absence of such modules in our model. We will change our discussion to give a better assessment of all factors affecting the achieved performance with complex data sets (e.g., lack of data augmentation) and how it relates to the state-of-the-art.

---

> > ### Comment · Reviewer_gTB4 · 2021-11-20
> > **Thanks for clarifying**
> >
> > Thanks for the clarifications.
> >
> > I am not sure I understand the theoretical memory footprint. It's clear why it's linear in C, but what exactly is "the resolution in each convolution layer after fitting" (N) referring to and why is memory linear in N?
> >
> > Also, it would be great to see an updated version of the manuscript with the promised revisions. I believe this is possible.

---

> > > ### Author Response · Authors · 2021-11-21
> > > **about the theoretical memory footprint**
> > >
> > > We will continuously update the revision, prioritizing the most critical changes first, so that reviewers can take a look at them while we are still in correspondence, until the deadline. Changes from the original manuscript will be highlighted in violet. For now, we have uploaded a revision in which we added an appendix for the theoretical memory footprint.
> > >
> > > There, we now explicitly define N_i = number of data Gaussians on the input, N_k = number of Gaussians in one kernel, F_i = number of input feature channels and F_o = number of outpt feature cannels. A convolution thus yields N_i * N_k * F_i Gaussians, which can be fitted inside the convolution operation to a smaller value N_o before forwarding them to the activation function module. In our previous answer, we wrote N instead of N_o and C instead of F_o, since these two factors contribute the most to the theoretical GMCN memory footprint. Its full definition, along with concrete examples, can be found in the new appendix.

---

> > > > ### Comment · Reviewer_gTB4 · 2021-11-25
> > > > **Still somewhat confused about dense fitting**
> > > >
> > > > Thanks for the update. I'm still a bit confused about the dense fitting step. Isn't the sign of a_i the product of the two mixture components' signs that generated it? The sign of the mixture component in the input feature map is positive per Eq. (6). If the kernel's component has a negative sign this component basically gets ignored (a_i = \epsilon). So doesn't this effectively mean that you might as well restrict the mixture components to have positive magnitudes in the first place?

---

> > > > > ### Author Response · Authors · 2021-11-25
> > > > > **About the dense fitting**
> > > > >
> > > > > That is an error that crept in while updating the notation as suggested by reviewer Ao7K. We decided to update all of the notation to the style used by Goodfellow et al. 2016, as seems the standard in this conference. Thanks for pointing that out.
> > > > >
> > > > > The correct version of the equation is \begin{align}
> > > > >     a''_i &= a'_i \times \frac{\varphi\left(\operatorname{gm}(\mathbf{B}_i, \mathbf{a}, \mathbf{B}, \mathbf{C})\right)}{\operatorname{gm}(\mathbf{B}_i, \mathbf{a'}, \mathbf{B}, \mathbf{C})},
> > > > > \end{align}
> > > > >
> > > > > In an old revision (https://openreview.net/references/pdf?id=VjgDB4gfSr) the equation is still correct. In the appendix it's also correct, since we did not update the notation style yet.
> > > > >
> > > > > We will definitely fix the error in Equation 7 and update the notation in the appendix before publishing the paper.

---

> > > > > > ### Comment · Reviewer_gTB4 · 2021-11-25
> > > > > > **Misunderstanding**
> > > > > >
> > > > > > Sorry I might have not been precise with my question: I wonder how a mixture component with a negative sign (in the kernel) will affect the result. in the result of the convolution, all resulting mixture components will have negative sign, since the activations are positive. The Eq 6 will set all those components‘ a‘s to \epsilon, i.e. make them positive and close to zero. So what’s the point of allowing negative weights in the first place?

---

> > > > > > > ### Author Response · Authors · 2021-11-25
> > > > > > > **effect of negative gaussians on the relu.**
> > > > > > >
> > > > > > > Ah, right.
> > > > > > >
> > > > > > > Yes, all these negative components will be close to zero in a'' as well. In the reduction step they will have small weights, and therefore they will not be considered during fitting almost at all.
> > > > > > >
> > > > > > > However, the negative components also influence the numerator through the sum of the gaussian mixture evaluation, and therefore the weight of other components in a''.
> > > > > > >
> > > > > > > For example:
> > > > > > > Imagine a mixture (before ReLU fitting) with 2 components centred around 0. one of them is -1, the other 2. Such a mixture could have resulted from convolution of a single Gaussian with a = (1), with a kernel with 2 Gaussians with a = (-1; 2). According to the equations, this will result in weights of around a'' = (ε;1).
> > > > > > >
> > > > > > > When computing the gradient during backpropagation, it will flow through a' only to the positive weights. but it will flow through the numerator to both, positive and negative, weights.

---

### Official Review · Reviewer_Ao7K · 2021-11-14

**Correctness:** 3
**Technical Novelty And Significance:** 3
**Empirical Novelty And Significance:** 3
**Recommendation:** 5
**Confidence:** 3

**Main Review:**

Strength: This paper provides a new solution to handle the high-dimensional problems.

Weakness:

1.   Compared with other methods, such as PointNet and PointNet++, the effectiveness and practicability of this method cannot be sufficiently demonstrated. It is recommended to validate GMCN on more complex datasets and to give a more adequate analysis.
2.   TreeHEM is used to reduce the number of Gaussians in the mixture, and  $T=2$ works best in terms of computation time, memory consumption, and often fitting error. The determination of hyper-parameter or specific values are based on experience.
3. The evaluation is quite insufficient as this paper claims that it is a general purpose deep learning The results are not very convincing on complex tasks.

Questions:
1.   In Equation 4, there are $N\times M$ terms, each term is $g_i(x)* g_j(x)$, so according to Equation 5, is the sum of each column substituted in the Relu function? In Equation 7, $B_i$ is unknown? $C$ in Equation 1 and $B$ in Equation 2 should be in bold form.
2.   This paper mentions that "The input GM is then processed by four consecutive blocks of Gaussian convolution  and ReLU fitting layers, where each doubles the number of feature channels and halves the number of Gaussians". So is the first layer $4\times 8$, and the  last layer $64\times 128$, with $4$ Gaussians?

**Summary Of The Paper:**

For the curse of dimensionality problem of CNNs, this paper presents the Gaussian mixture convolution network (GMCN), a method that alleviate the problem of high-dimension data. GMCN is a deep functional network method that represents data as functions. It fits a Gaussian mixture to result of transfer functions, such as RELUs. The effectiveness of this architecture is verified on MNIST and ModelNet10.

**Summary Of The Review:**

The method presented in this paper is new, but more detailed analysis and evaluation are needed to make this work more solid.

---

> ### Author Response · Authors · 2021-11-20
> **Answer to Reviewer Ao7K**
>
> We thank the reviewer for their review and the time spent assessing our submission. We will address the raised questions below.
>
> >Concerns regarding the evaluation
>
> We agree that a more extensive evaluation would be optimal; however, as the reviewer correctly pointed out, we present a general-purpose method for deep learning. Providing an adequate impression over all possible use cases, along with the exposition, is challenging, especially given the rather compact format requirements. In its current form, we tried to focus on the results where our approach reaches state-of-the-art performance. However, we would be happy to provide additional results and insights for ModelNet40, which could be introduced to the evaluation or the paper’s appendix.
>
> These would include experiments where our approach can surpass established convolution-based baselines but occasionally falls short of state-of-the-art performance. In accordance with reviewer gTB4, we would further add comparisons against relevant voxel-based approaches on the tested data sets. As a quick preview of these results, on ModelNet10 we have 93.3% accuracy, while VoxNet achieves 92% and OctNet 91.5%. On ModelNet40 we have 87.6%, while VoxNet is at 83%, O-CNN at 90%, and NormalNet Vanilla at 87.7%, making our approach competitive with these prior, convolution-based baselines. We would further like to highlight that, due to the limited performance of the current prototype implementation, we did not apply data augmentation (in contrast to O-CNN). We also did not use additional features (like normals in NormalNet). Other, more recent approaches may perform even better on these data sets, but they are usually not 3D convolution-based. Instead, they are view-based (e.g., iMHL and MLVCNN), or include purpose-built feature-learning modules in the network (e.g., relation-shape in RS-CNN or reflection-convolution-concatenation in NormalNet RCC). Such purpose-built modules are feasible with our architecture but out of scope for this paper.
>
> ---
>
> >Selection parameters for TreeHEM
>
> We did experiments on the T parameter and described this briefly in Section 5.3, so the choice is based on experiments rather than experience. In the revised version, we will explicitly state in the method’s description that experiments can be found in the evaluation in Section 5.3. However, we agree that the comparison of only T=2 and T=4 is too limited, and we propose to include further results for T=8 in the final manuscript (obtaining these will take time) to highlight the corresponding trends and that T=2 is the best choice. We will also add a suitable explanation for these trends. Performance and memory footprint are easily explained since we cache T Gaussians in each tree node, and therefore memory access costs are linear with respect to T. Regarding lowered accuracy, we reckon that it can be attributed to the smearing of Gaussians (i.e., loss of fine details), which is more likely to occur when more of them are merged simultaneously.
>
> ---
>
> >Question regarding equations 4, 5, and 7
>
> The indices in Eq. 4 and 5 are unrelated, i.e., the ReLU has to run through all NxM Gaussians that come out of the convolution. We will switch to different index letters to make this more apparent in the revision. In Equation 7, only a' ' is unknown. Regarding the notation in all equations, we will switch to the recommended notation from the template (Goodfellow et al. 2016) in the new revision.
>
> ---
>
> >Number of Gaussian in each layer
>
> Regarding the convolution layers, we start with one feature channel (density), so the first layer is 1x8. The second-to-last layer is 32x64 with 8 Gaussians, and the last layer is 64x10 without reduction (as shown in Figure 4). We will try to make this clearer.

---

### Author Response · Authors · 2021-11-23
**Changed revision uploaded**

We thank the reviewers for their detailed and constructive feedback. We have now finished integrating the majority of the requested changes into our manuscript and uploaded it for your consideration. The relevant portions that were modified or newly added (mostly appendix content) are marked by purple font color. We look forward to your assessment and continued discussion.

---

### Decision · Program_Chairs · 2022-01-20

**Decision:**

Accept (Poster)

**Comment:**

This paper presents a deep learning method that aims to address the curse-of-dimensionality problem of conventional convolutional neural networks (CNNs) by representing data and kernels with unconstrained ‘mixtures’ of Gaussians and exploiting the analytical form of the convolution of multidimensional Gaussian mixtures. Since the number of mixture components rapidly increases from layer to layer (after convolution) and common activation functions such as ReLU do not preserve the Gaussian Mixtures (GM), the paper proposes a fitting stage that fits a GM to the output of the transfer function and uses a heuristic to reduce the number of mixture components. Experiments are presented on MNIST (2d) and ModelNet10 (3D), which show competitive performance compared to other approaches such as classic CNNs, PointNet and PontNet++ methods.

There is somewhat an overall consensus on the novelty of the proposed approach and its potential to pave the way for further research. There were, however, several issues raised by the reviewers in terms of clarity, memory footprint and computational cost that limits the applicability of the method to more complex datasets. While the authors expanded on the dense fitting in their comments and in the revised version of the paper, it still remains unclear the role of the negative weights, as the dense fitting stage seems to constrain all the weights to be positive. In terms of memory footprint, the authors refer to the theoretical footprint and their implementation does not match this. Finally, it is acknowledged by the authors that the computational cost is a limitation that hinders the method from achieving competitive performance in more complex tasks.